**communications** engineering

# Optimising housing typology distributions for multi-hazard loss reductions in resource-constrained settings
Arvin Hadlos ✉, Aaron Opdyke & S.Ali Hadigheh

Disaster loss estimations are valuable risk reduction tools but rarely consider the loss trade-offs when a building stock is subjected to multi-hazard impacts. Here, we developed an approach to simulate direct economic losses of a housing stock and explore loss reduction across scenarios of housing typology distributions. We used multi-objective optimisation to model wind and seismic losses in Itbayat, Batanes, Philippines. Using Monte Carlo simulation, 11,628 housing stock scenarios were modelled under two cases of paired extreme hazard intensity thresholds, identifying Pareto optimal solutions that were further analysed against a socio-technical framework. We show that the current housing stock distribution can sustain lower multi-hazard losses by achieving more optimal combinations of lightweight and reinforced concrete typologies. However, transitioning to this desired stock distribution becomes a trade-off of not just wind-seismic loss reductions but also of socio-technical considerations such as households' risk perceptions. Our study advances risk reduction strategies by streamlining loss estimations to inform collective and safer multi-hazard construction practices.

Disaster losses serve as an important metric to assess the toll that natural hazards take on societies. In 2023, global disaster losses were estimated to be USD 250 billion[1,2], conveying the extensive economic impacts of disaster events worldwide. A sub-component of disaster losses is direct economic loss, which represents the monetary consequences of natural hazards inflicting damage to physical assets (e.g., infrastructures and buildings)[3,4]. Hence, direct economic loss is crucial in understanding the value or cost of structural repairs, rebuilding efforts or required investments to mitigate potential losses. In the United Nations Sendai Framework for Disaster Risk Reduction, the reduction of direct economic losses is among the seven global targets contributing to the overall goal of substantially reducing global disaster risks[4].

In 2023, the series of earthquakes in Türkiye and Syria[5] was the single most impactful disaster event, causing USD 50 billion in damages[1,2]. However, the majority of direct economic losses (76%) in that year were from weather events such as typhoons and hurricanes[1,2]. While some natural hazards can be less frequent and have high impacts, others accumulate losses over their frequent occurrence. Many communities globally are exposed to multi-hazard impacts which complicate strategies to prevent property damage. "Multi-hazard" can be defined as the presence of at least two hazards that overlap spatially or temporally, either with or without dependence on each other[6–9]. In this study, we focus on the spatial overlap of natural hazards over a geographical location, implying the exposure of

communities to at least two independently occurring hazards. In the Philippines, for example, 60% of the country's total land area experiences multiple hazards (such as typhoons and earthquakes), and 74% of its population is vulnerable to their impacts[10]. Often, the competing impacts of hazards mean that addressing one hazard may increase the vulnerability of assets to the other and vice versa[11,12]. Identifying optimal solutions is thus foundational to inform strategies to simultaneously address impacts from two or more hazards that often have diverging requirements.

Traditionally, disaster risk reduction (DRR) has long relied on a single-hazard approach where hazard impacts are assessed or analysed in isolation from other hazards[13]. The attention to understanding multi-hazard impacts has gained traction, acknowledging their compounding consequences for communities[14,15]. Of the 16,535 global disaster records from 1900 to 2023, 19% can be (re)classified as multi-hazard events (considering the spatial or temporal overlaps of hazards), constituting 59% of total economic losses[16]. In addressing multi-hazard impacts, necessary trade-offs are required as a compromise to satisfy diverging and conflicting requirements. For example, in past research analysing building-level risk reduction against both floods and earthquakes in Afghanistan, optimal structural measures were found to vary spatially in that some districts could reduce disaster risk by investing more in flood measures compared to seismic mitigation, and vice versa[11]. The concept of optimality thus emerges as a necessary strategy to analyse trade-offs amid conflicting requirements of reducing disaster risks.

School of Civil Engineering, The University of Sydney, Sydney, NSW, Australia. ✉e-mail: arvin.hadlos@sydney.edu.au

**Fig. 1 | Predominant housing typologies in Itbayat, Batanes, Philippines. a** A lightweight typology that can either have timber posts and beams (LWA) or steel posts and timber beams (LWB). **b** A semi-concrete typology that can either have steel posts and timber beams (SCA) or reinforced concrete (RC) posts and timber beams (SCB). **c** An RC typology with lightweight roof (RCA). **d** An RC typology with RC slab roof (RCB).

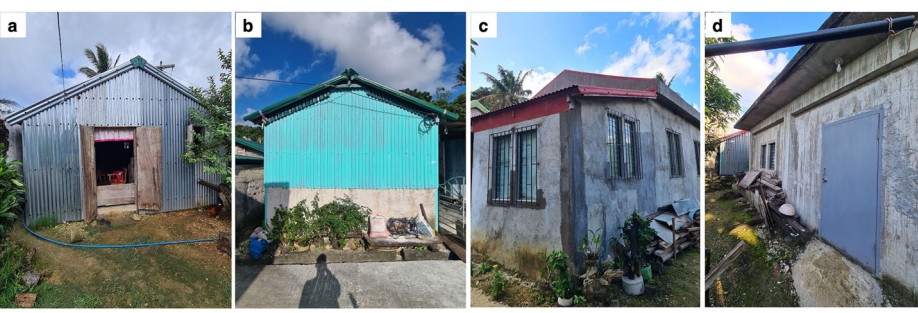

Adherence to a single-hazard approach in a multi-hazard context is insufficient for holistic risk reduction and is counterintuitive. Due to the differing dynamics of counteracting multiple hazards, investments towards structural risk reduction against only one hazard can create risk to other hazards.

Present guidance instruments to reduce structural risks, such as building codes, are mostly tailored to a single hazard outlook. The design of structures is (usually) governed by the effects of the hazard that dominates[17,18]. This approach overtly assumes that satisfying the load demands of the dominant hazard also addresses the load requirements of its non-dominant counterpart/s. However, this assumption can be problematic. Structures subjected to different loads, such as seismic and wind loads, can potentially have up to twice the risk of exceeding limit states compared to only considering risks associated with a single hazard that dominates, as is usually prescribed in code provisions[19]. The need to understand the synergy in multi-hazard design[20] thus emerges, motivating a need to acknowledge structural impacts from all known hazards, whether they dominate or not[17,18,21]. This is critical in multi-hazard settings where structural risks are being complicated by the exposure of assets to independently occurring hazards having different dynamics, frequencies, and impacts, obscuring straightforward pathways towards risk reduction.

An integrated analysis of the impacts of spatially overlapping natural hazards has implications for a more holistic understanding of their collective consequences to physical assets eventually influencing construction decisions in a specific geographical area. Multi-objective optimisation is well established in the field of engineering. Still, its applications for multi-hazard analysis have primarily been concentrated on the trade-offs pertinent to specific structural components[18,21,22] or at an individual building level[23,24]. Community-wide applications in past studies have used multi-objective optimisation to model the trade-offs among direct economic loss, population dislocation[25], and post-disaster repair times[26]. So far, multi-objective optimisation has yet to be capitalised to analyse the trade-offs of structural impacts to physical assets at a community level (e.g., a building stock) against multiple hazards. Addressing this gap has critical potential to transform how we reduce risk especially in building stocks with highly variable (or "heterogeneous") physical assets. This larger unit of analysis is poised to support community efforts towards risk reduction through the potential regulation of structural typologies in heterogeneous building stocks susceptible to multi-hazard exposure. While DRR efforts have focused on the repair and retrofit of (existing) structural assets as practical solutions to strengthen physical assets, these measures can be insufficient considering that structural characteristics of typologies can govern vulnerability[27,28]. Within informal construction markets, addressing deeply seated vulnerabilities to multiple hazards requires key changes which we position can be more practically achieved through changes to building typologies. Thus, in a heterogeneous building stock, there is an opportunity to achieve optimal combinations of different building typologies to minimise multi-hazard impacts to safeguard the collective structural assets of a community.

Localised disaster events experienced by remote and rural communities contribute remarkably to disaster losses, but these often remain unreported because they do not generate attention on national and global scales[29]. From 1990 to 2013, 99.7% of all disasters were localised or small-scale events, constituting less than 30 deaths and less than 5000 houses destroyed[30]. Disaster data – and presumably risk analysis – has been biased towards high-impact disaster events, often in urban areas and regions with higher asset values where "reportable" disaster loss values are larger[29,31,32]. In resource-constrained settings typical in low- and middle-income countries, dwellings tend to be non-engineered houses built informally by households or local construction workers without oversight from built environment professionals such as engineers and architects[33]. A heterogeneous housing stock emerges in such an instance characterised by a high variance of housing typologies. In this case, regulating safer construction practices becomes challenging and usual repair and retrofit strategies have limitations because the characteristics of housing typologies can become the main drivers of vulnerability to hazards. Thus, a broader and more collective view of understanding multi-hazard impacts of housing assets on a community scale is crucial for more encompassing prevention of damage to assets. This study focused on the context of a municipal-level assessment of direct economic losses of a housing stock in a geographically remote and small island community. Factoring in the intricacies of navigating the competing impacts of wind and seismic hazards and acknowledging the intrinsic characteristics of the local residential portfolio, we ask the question: How can a heterogeneous housing stock be optimised against multi-hazard direct economic losses?

The main objective of this study is to explore transitions towards more optimal housing stock distributions which simultaneously minimise direct economic losses from both wind and seismic hazards. Mitigating the impacts of both hazards usually requires trade-offs because of their potentially incongruent structural requirements. For example, ductile performance is essential for a structure in seismic regions but this is less critical for wind load design[17]. This trade-off requires Pareto optimal solutions[34] representing candidate solutions that somehow satisfy the conflicting requirements of various objectives. Our study is based in Itbayat, Batanes – a frequent typhoon passage in the northern Philippines. The exposure of this municipality to wind impacts has greatly influenced the historical construction of heavy, masonry dwellings[35,36] but these were destroyed after a series of earthquakes in 2019[37,38]. New housing typologies emerged thereafter[39,40] but with limited knowledge of their wind and seismic performance. We focused on wind and seismic hazards because they are the most prominent hazards and the only known hazards that inflict notable damage to housing assets in the selected municipality. This was based on their Comprehensive Land Use Plan and on the in-person consultations in 2023 with the Municipal Planning and Development Office and the Municipal Engineering Office.

For this study, we explored different hypothetical housing stock distributions (referred to as "scenarios") to identify what concentrations and ratios of the emergent housing typologies can minimise multi-hazard direct economic losses. These typologies are described in Fig. 1 and Table 1. We generated 11,628 housing stock scenarios based on the assumption that for a given housing stock, each typology would constitute at least 5% of the overall stock distribution. We then simulated the loss outputs for each scenario

**Table 1 | Characteristics of the different housing typologies**

| Housing typology | Characteristics |
|---|---|
| Lightweight with wooden posts (LWA) | One story; timber beams and columns (no footings); corrugated galvanised iron (CGI) sheets as building envelopes; gable roof profile |
| Lightweight with steel posts (LWB) | One story; 4-inch to 5-inch diameter steel pipes as posts (with reinforced concrete (RC) footings) and timber beams; CGI sheets as building envelopes; gable roof profile |
| Semi-concrete with steel posts (SCA) | One story; 4-inch to 5-inch diameter steel pipes as posts (with RC footings) and timber beams; half-height RC walls at the base with remaining CGI walls for the upper half; gable roof profile with CGI sheets |
| Semi-concrete with reinforced concrete posts (SCB) | One story; RC posts (with RC footings) and timber beams; half-height RC walls at the base with remaining CGI walls for the upper half; gable roof profile with CGI sheets |
| Reinforced concrete with lightweight roof (RCA) | One story; RC posts (with RC footings) and beams; RC walls; gable roof profile with CGI sheets |
| Reinforced concrete with slab roof (RCB) | One story; RC posts (with RC footings) and beams; RC walls; RC slab roof |

For detailed structural descriptions such as post-to-beam connections, see Hadlos et al.[39].

using Monte Carlo simulation, considering two cases of paired extreme hazard intensity thresholds. Seismic intensities are based on the Philippine Volcanology and Seismology Earthquake Intensity Scale (PEIS) which is the nationally developed earthquake intensity measurement in the Philippines[41]. PEIS range from Intensity I (scarcely perceptible; approximate peak ground acceleration (PGA) of <0.0005 g; equivalent to Modified Mercalli Intensity (MMI) I) to Intensity X (completely devastating; approximate PGA of >1.39 g; equivalent to MMI XII)[41–44]. Meanwhile, wind intensities are expressed in kilometres per hour (km/h) and represent a 3-second peak gust measured 10 m above ground[45]. The first case of our analysis represents occasional hazard occurrences (PEIS VII and 270 km/h) pertaining to hazard levels below the maximum projected thresholds. The second case is rare hazard occurrences (PEIS VIII and 300 km/h) which represents the projected maximum hazard intensities. For both cases, Pareto optimal solutions were identified and then ranked using a multi-attribute decision-making method called the R-method[46]. Finally, acknowledging the socio-technical factors influencing residential construction in resource-constrained settings, the Pareto optimal solutions were contextualised against the desirability, viability, and feasibility (DVF) framework. We adopted this framework to qualitatively discuss whether potential solutions align with the households' preferred mode of construction ("desirability"), suitable for the community in the long term ("viability"), and realistically achievable ("feasibility"). We assessed the DVF criteria based on field study insights of housing reconstruction trajectories of the households in Itbayat. For the entire data analysis procedures of this study, see "Methods" section.

Our main findings suggest that the current housing stock distribution can be improved to sustain lower wind and seismic losses by achieving a more optimal combination of lightweight and reinforced concrete typologies at varying proportions within the building stock. This result instils a paradigm shift in understanding structural safety as an aggregate system of the physical assets of disaster-affected communities, safeguarding these assets more collectively. However, transitioning to this desired stock distribution becomes a trade-off of not just loss reduction between the two natural hazards but, more importantly, of socio-technical factors such as households' risk perceptions, local appropriateness of solutions, and availability of resources. Our study contributes to advancing risk reduction strategies by offering an approach that not only streamlines multi-hazard loss estimations and trade-offs but also contextualises these based on the capacities of constituents. This approach is useful for policymakers to inform collective safer construction practices in multi-hazard settings.

## Results
### Case 1 (Occasional hazard occurrences) – PEIS VII and 270 km/h
Sixty-five (65) Pareto optimal solutions were identified from the 11,628 scenarios for Case 1 (see Fig. 2; refer to Supplementary Table S7 in the Supplementary Material for the list of Pareto optimal solutions). The two top-ranked solutions (tied at first rank) based on the R-method lie on the

extreme ends of the Pareto frontier. This means that either a housing stock dominated by 75% LWB or 75% RCB is the most optimal solution to reduce losses among the rest of the non-dominated solutions. However, such a result is heavily based on the "polarity" of the impacts of wind and seismic hazards made pronounced when we specified equal ranks for each hazard. The polarity infers that heavier typologies will incur larger losses to seismic, and lighter typologies will have more losses to wind impacts. This yielded extreme results where a tandem of losses of PHP 331 M for seismic and PHP 195 M for wind was identified as an equally superior solution alongside losses of PHP 162 M for seismic and PHP 385 M for wind. Practically, between these two solutions, there is no merit in picking one over the other as it would substantially skew loss prevention to either hazard. Thus, alternatives can be explored along the Pareto frontier which strike the balance of minimising losses for both hazards.

The baseline scenario or the current housing stock distribution in the community is characterised by a spread of 10–25% of each typology, with an estimated loss of PHP 199 M for seismic and PHP 382 M for wind (see Fig. 3). The current baseline is distant from the Pareto frontier, confirming that shifts in the distribution of housing typologies would result in improvement to minimise losses for both hazards. The closest Pareto optimal solution from the baseline is the most immediate option along the Pareto frontier which reduces the losses against both hazards from the current baseline losses. This scenario constitutes a majority of lightweight structures (50% LWA & 20% LWB) and an increase to 15% fully concrete houses (RCB), with the remainder of typologies at 5% each. This scenario offers a slight reduction in seismic losses at PHP 191 M and a notable decrease in wind losses at PHP 330 M (see Fig. 3). Along the Pareto frontier, a scenario with balanced losses for both hazards is a housing stock dominated by a combination of 40% LWA and 40% RCB (with the rest of the typologies at 5% each). The losses for this scenario are estimated to be PHP 251 M for seismic and PHP 269 M for wind. However, moving from the baseline scenario to here would drastically exacerbate seismic losses but will notably reduce wind losses.

The transition from the baseline scenario is not only a concern of the extent to which losses are reduced but also of the socio-technical factors necessary to support and manage such transition. Adopting the 75% LWB distribution considerably reduces seismic losses and offers almost no reduction to wind losses, and this scenario compromises the desirability criteria in that concrete-based structures are heavily favoured by the community more than lightweight houses. Additionally, while this scenario is feasible cost-wise, the social acceptance of these as permanent dwellings needs to be factored in with additional attention to how their longevity could be improved given that these tend to be less durable than concrete structures. On the other hand, the 75% RCB distribution is desirable and viable, but the transition would not be financially feasible, and it only reduces losses largely to wind and greatly exacerbates losses to seismic. The 40% LWA and 40% RCB distribution is potentially the best candidate solution that balances the

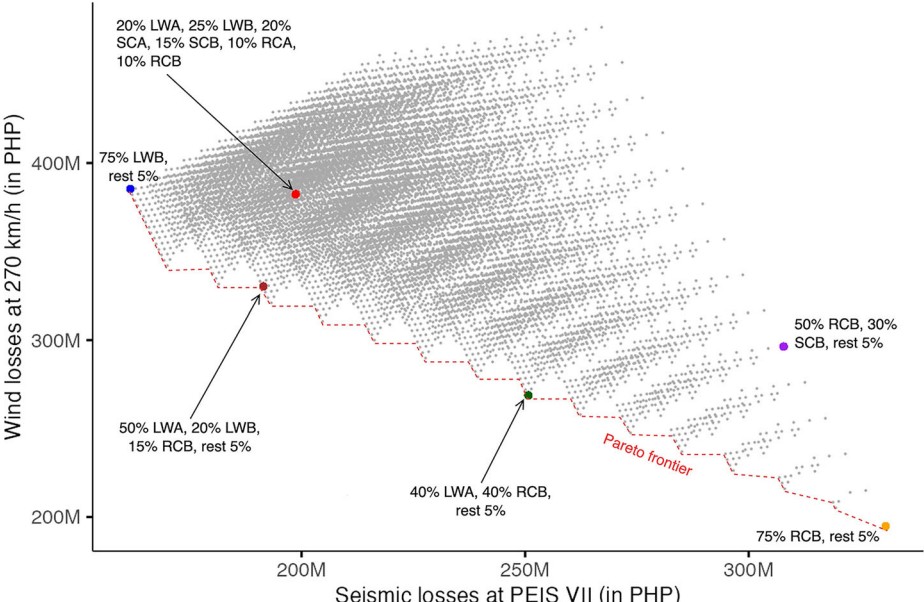

**Fig. 2 | Pareto optimal graph for Case 1.** The grey points represent the 11,628 scenarios simulated for seismic and wind losses at PEIS VII and 270 km/h, respectively. The red, broken line shows the Pareto frontier where the non-dominated (Pareto optimal) solutions lie. The superimposed coloured points show the important scenarios. The red point is the baseline representing the current housing stock distribution. The blue and orange points are the first-ranked solutions. The brown point is the solution closest to the baseline. The green point is the solution that yields the most balanced minimal losses for both hazards while the purple point represents the most balanced maximal losses. (Note that the values are based on the 95th percentile of the resulting probability distribution for each scenario. Losses are in Philippine peso (PHP). PEIS refers to the Philippine Volcanology and Seismology Earthquake Intensity Scale. LWA refers to lightweight typology with timber posts and beams; LWB for lightweight typology with steel posts and timber beams; SCA for semi-concrete typology with steel posts and timber beams; SCB for semi-concrete typology with reinforced concrete (RC) posts and timber beams; RCA for RC typology with lightweight roof; and RCB for RC typology with RC slab roof).

conflicting demands of the desirability, viability, and feasibility criteria although with a compromise in preventing seismic losses considering the baseline. While the closest optimal solution to the baseline might not be desirable given that it is predominantly lightweight, this scenario is the most practical trajectory to lessen multi-hazard losses.

### Case 2 (Rare hazard occurrences) – PEIS VIII and 300 km/h

Fifteen (15) Pareto optimal solutions were identified from the 11,628 scenarios for Case 2 (see Fig. 4; refer to Supplementary Table S8 in the Supplementary Material for the list of Pareto optimal solutions). Similar to Case 1, the two top-ranked solutions (tied at first ranks) based on the R-method lie on the extreme ends of the Pareto frontier, reinforcing the polarity of wind and seismic impacts in a heterogeneous housing stock comprised of either mostly lightweight or mostly concrete typologies. The first-ranked solutions estimate PHP 629 M losses for seismic and PHP 331 M for wind if the distribution is 75% RCB; PHP 311 M losses are estimated for seismic and PHP 403 M for wind if the distribution is 75% LWA. Direct economic loss estimates are partly dependent on construction costs. When considering a heterogenous housing stock under extraordinary hazard intensities, it follows that more expensive typologies will incur more losses while the least expensive ones incur fewer. This is under the assumption that extreme hazard intensities overthrow the disparate housing performance among the typologies. Under such circumstances of extreme events, mitigation strategies should at least aim for collapse prevention to reduce casualties, problematising at what level of construction investment is commensurate to adhere to this housing performance in resource-constraint settings.

The baseline scenario – of which the typologies are spread within 10% to 25% distributions – yields PHP 386 M losses for seismic and PHP 474 M for wind (see Fig. 5). Losses can be further minimised from the baseline by transitioning to a scenario with 70% LWA, 10% RCB, and the rest of the typologies at 5% each. With such a transition, the reduction of losses is estimated at PHP 334 M for seismic and PHP 398 M for wind (see Fig. 5). This is the most ideal trajectory when considering notable improvements to

both losses. Alternatively, when opting for balanced losses for both hazards, transitioning to a housing stock with 60% LWA, 20% RCB, and rest at 5% each would incur losses of PHP 380 M and PHP 388 M for seismic and wind, respectively. This trajectory also reduces losses for both hazards compared to the baseline, although it is more remarkably towards wind loss reduction and not so for seismic.

Similar to Case 1, the Pareto solutions on the extreme ends of the Pareto frontier are likely neither the most pragmatic solutions in that they highly prevent losses for one hazard but to a notable detriment of loss reduction to the other hazard. When factoring in the DVF criteria, their extremities are further stretched considering that one becomes a highly desirable and viable option but not feasible, and vice versa. To this end, the most practical trajectories are the transitions that improve loss reduction for both hazards compared to the baseline, albeit to the compromise of some of the criteria in the DVF framework. For example, the substantial loss reduction to both hazards by transitioning to either 70% or 60% LWA is promising to manage multi-hazard impacts but raises potential conflict with the desirability and viability criteria. Transitioning to a housing stock with more lightweight structures is more plausible given that financial limitations might override the aspirational aspects of desired construction trajectories among households. However, the technical standpoint of endorsing lightweight typologies needs further evaluation in a storm-battered region with more frequent wind events than seismic activities.

### Discussion

The housing stock-level analysis of optimising typologies within a building portfolio against multi-hazard losses, as demonstrated in this study, offers a new way to approach structural risk reduction. This analysis is crucial in settings like Itbayat where "experimental" and non-engineered housing typologies emerged due to the immediate need for households to shelter themselves using their available resources following disaster impacts. The heterogeneity of the housing stock – arising from the plurality of how households constructed their dwellings – poses challenges for individual

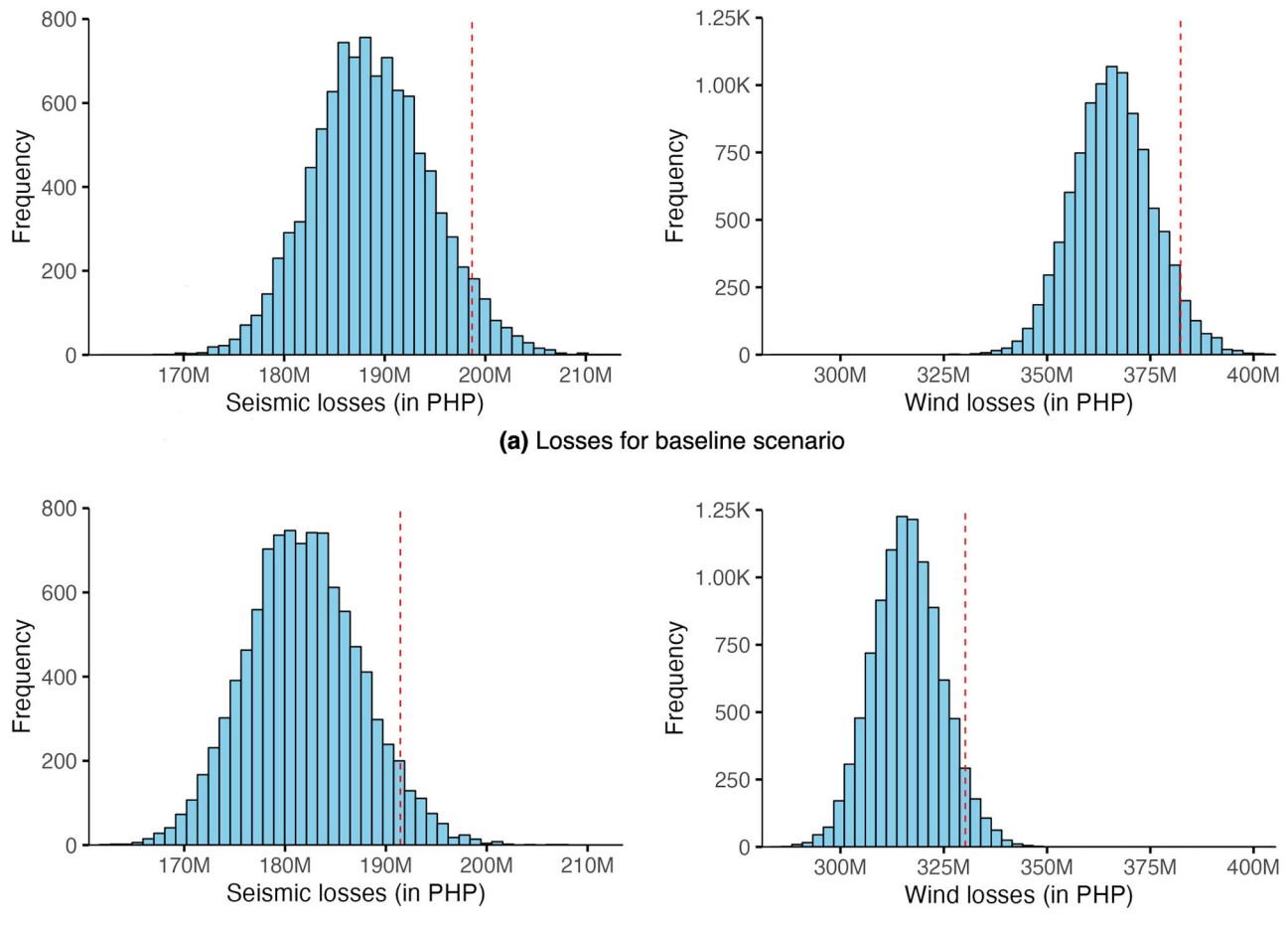

**(a)** Losses for baseline scenario

**(b)** Losses for the Pareto optimal solution closest to the baseline scenario

**Fig. 3 | Simulated losses for Case 1 (PEIS VII and 270 km/h). a** The baseline scenario and **b** the Pareto optimal solution closest to the baseline scenario. The red, broken line indicates the 95th percentile value. (Note that PEIS refers to the Philippine Volcanology and Seismology Earthquake Intensity Scale).

structural housing risk mitigation. This circumstance necessitates a more collective appraisal of the characteristics of the housing stock in the community to leverage substantial risk reduction efforts. Under very high to extreme hazard intensities, usual repair and retrofit schemes can have limitations in addressing structural safety, and construction of or adherence to specific typologies might offer more effective pathways to reduce hazard impacts. Therefore, the stock-level methodology of optimising housing typology distribution used in this study aims to streamline robust and wider risk assessment beyond the individual housing level. This methodology can provide community-level guidance into strategies to mitigate structural losses by increasing or decreasing concentrations of certain typologies which can be achieved by the entry points discussed below. The endorsement of optimised housing typology distributions not only yields anticipated lower economic losses in the event of disasters but will also potentially decrease the number of casualties and reduce the length of repair times for impacted assets.

Our application of multi-objective optimisation to a collective approach for structural risk reduction complements and further substantiates insights into engineering optimisation models currently concentrated on building-level components of assets[18,21–24]. Given the escalating global impacts of multi-hazard events that complicate DRR efforts, a paradigm shift is needed to understand structural safety as an aggregate system of the physical assets of disaster-affected communities. If, for example, a housing stock is perceived as an aggregate system and not just a composition of individual dwellings, the implications can be far-reaching. These include instigating equitable distribution of building resources in a socio-demographically diverse community to having a more cohesive

outlook in planning agendas (e.g., zoning reforms) in multi-hazard geographies. Our study has also highlighted – and challenged – the concept of optimality in resource-constrained settings. With the notion of optimal solutions in engineering often limited to the knowledge of material performance trade-offs of assets, we extended our analysis to account for the socio-technical factors that explain the social realities impacting realistic pathways to prevent multi-hazard losses. Thus, our approach adds a layer to an otherwise tangible view of trade-off analysis, guiding future applications in the pragmatic assessment of disaster losses for similar contexts.

The housing stock transitions proposed in this study can be implemented through the variable increase and decrease of housing typology quantities (based on the prescribed ratio of housing stock combinations). This is a longitudinal process due to the dynamic housing needs influenced by the (expected) demographic changes over time, such as shifts in population, household sizes, and living setups or preferences. As such, this is a sustained effort that can be championed by planning and engineering officials to work towards optimal stock distributions within their jurisdictions as a form of disaster mitigation. Increasing certain typology quantities means encouraging new construction or incremental housing modifications towards desired typologies, all while controlling or inhibiting the construction of other typologies (thus "decreasing" their ratios relative to the uptake of the new construction or housing modifications). In practice, a starting point for this to happen is through having community-wide construction guidelines enforceable through development approvals and permitting processes. It must be noted that housing stock transitions in resource-constrained settings are not straightforward processes[47] because construction assets might not be accrued easily due to socio-demographic

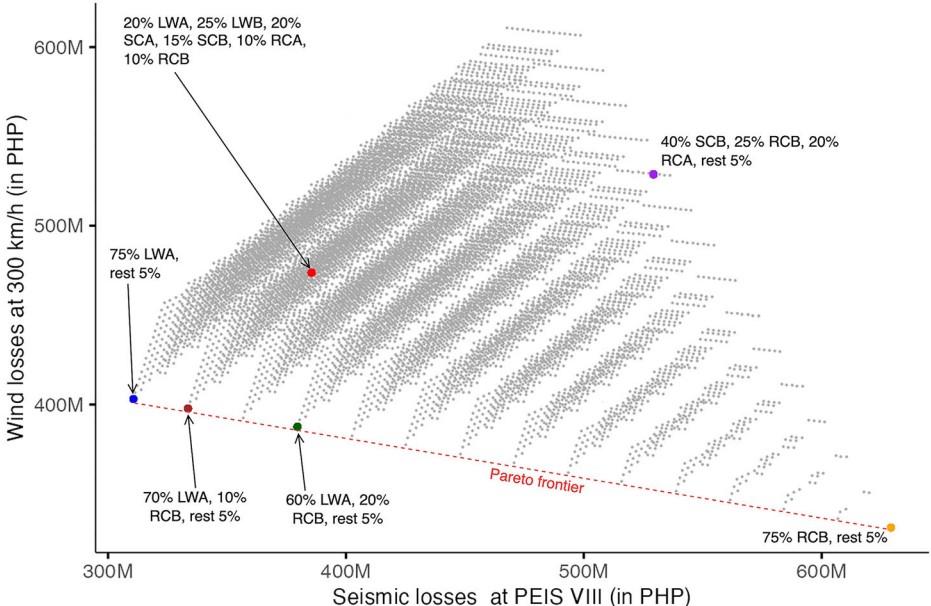

**Fig. 4 | Pareto optimal graph for Case 2.** The grey points represent the 11,628 scenarios simulated for seismic and wind losses at PEIS VIII and 300 km/h, respectively. The red, broken line shows the Pareto frontier where the non-dominated (Pareto optimal) solutions lie. The superimposed coloured points show the important scenarios. The red point is the baseline representing the current housing stock distribution. The blue and orange points are the first-ranked solutions. The brown point is the solution closest to the baseline. The green point is the solution that yields the most balanced minimal losses for both hazards while the purple point represents the most balanced maximal losses. (Note that the values are based on the 95th percentile of the resulting probability distribution for each scenario. Losses are in Philippine peso (PHP). PEIS refers to the Philippine Volcanology and Seismology Earthquake Intensity Scale. LWA refers to lightweight typology with timber posts and beams; LWB for lightweight typology with steel posts and timber beams; SCA for semi-concrete typology with steel posts and timber beams; SCB for semi-concrete typology with reinforced concrete (RC) posts and timber beams; RCA for RC typology with lightweight roof; and RCB for RC typology with RC slab roof).

circumstances. Below, we outline some practical strategies for housing transitions to take place realistically.

In the context of new construction, the transition to the ideal ratios of housing stock distributions can be achieved through the following entry points. First, the construction of better-performing typologies can be encouraged by providing incentives to households intending to build dwellings (either in post-disaster settings or under normal situations). In areas like Itbayat where financial capacities greatly influence the typology households plan to build, the availability of "template" construction drawings could be a starting point to mainstream their adoption. Not only will this reduce the costs for engineering and architectural services, but it will also potentially embed the community-wide practice of adhering to endorsed construction specifications for such typologies. In the long term, this can then become the status quo of construction within the locale, with previous research showing that households can easily conform to construction trends to ultimately decide/aspire for how they will build their house[48,49]. These recommendations can inform municipal shelter recovery plans as promoted in the Post-disaster Shelter Recovery Policy Framework of the Philippines[50] or could be additionally enforced through the regulation of building permit approvals. In anticipation of the New Philippine Building Act that would allow municipal governments to legislate their own building laws[51], these recommendations can then be formally enacted through local building ordinances.

Second, in times of disaster, donors aiming to provide shelter kits and construction materials to impacted households can tailor their donations to the materials needed to construct targeted housing typologies. This also includes designing trainings contextualised locally[49]. For example, in Itbayat, the community has long practised timber construction as part of their cultural ways of house-making, but the adoption of concrete construction needs further technical guidance to avoid common defects (e.g., honeycomb). Donor agencies and organisations can design trainings appropriate in such instances to carry forward both familiar and foreign construction methodologies in housing (re)construction.

Housing stock transition considering existing housing assets can be accomplished through incremental upgrades as realistic pathways to achieve ideal concentrations of housing typologies within a community. For example, a semi-concrete house can be developed over time into a fully concrete structure given the suitable materials and expertise. Incremental upgrades should, however, be approached with caution to avoid creating new vulnerabilities that can arise through mixing materials (e.g., combining masonry infill walls with timber-framed systems). A fully concrete house will most likely not be downgraded to a lighter-weight version given the financial investments made and its desirability in the setting we have studied. The incorporation of lightweight typologies within the general housing stock distribution is then only possible with new construction to adhere to the ideal ratios. This may be feasible given how lightweight typologies are the least expensive to construct, and while they are perceived to be less desirable, it is more likely the case that financial feasibility is the main determinant driving construction in remote settings. Given the frequency of wind events in the region, the endorsement of lightweight typologies might seem counterintuitive, but ways to safeguard these assets have traditionally been practised in the area (e.g., tying of houses)[40]. Additionally, while the results of our simulation yield specific ratios among the typologies, further insights can be explored to understand whether the nuances of the structural characteristics between similarly classed typologies (e.g., LWA vs LWB for the lightweight class) have commensurate differences in their housing performance to that of their differential cost.

This study did not incorporate site exposure multipliers (e.g., hazard impacts on hilly locations, proximity to coastlines, soil-structure relationships) which can further potentially impact housing performance. However, our study sparks conversation on the importance of the spatial spread of housing typologies within a community and the role of land use planning as an instrument to reduce multi-hazard losses. Given the heterogeneity of the housing stock that assures some typologies are better off against wind than seismic impacts, and vice versa, there is a potential to regulate the construction of susceptible typologies in identified areas with higher exposure to

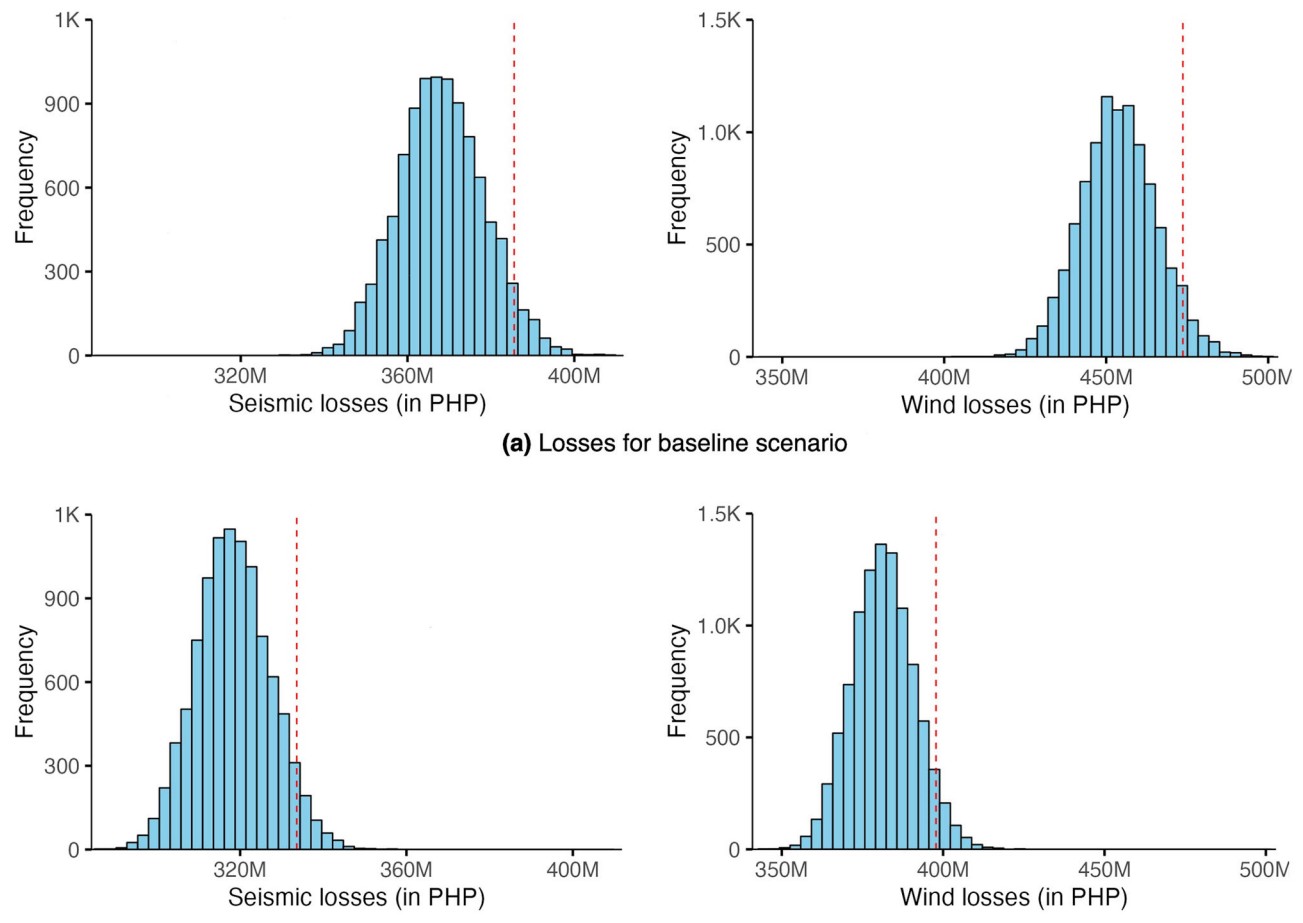

**(a)** Losses for baseline scenario

**(b)** Losses for the Pareto optimal solution closest to the baseline scenario

**Fig. 5 | Simulated losses for Case 2 (PEIS VIII and 300 km/h). a** The baseline scenario and **b** the Pareto optimal solution closest to the baseline scenario. The red, broken line indicates the 95th percentile value. (Note that PEIS refers to the Philippine Volcanology and Seismology Earthquake Intensity Scale).

a specific hazard. For example, no-build zones can be enforced where wind exposure is the strongest and seismic codes can be reviewed where they benefit soil-structure relationships while considering wind-seismic trade-offs accordingly towards decision-making. Additionally, the heterogeneity of the housing stock can be an advantage to intersperse typologies so that heavier typologies can act as wind buffers protecting lighter-weight counterparts. Our study focused only on the ratios of the housing stock distribution; thus, we envision our assessment to be the first layer of analysis with the most refined outputs possible once site exposure multipliers are considered.

Our study has several limitations. The direct economic loss estimation in this study is based on first-generation fragility functions[39] specifically derived for the field study site which used an expert opinion-based approach. Loss estimates can be refined if hybrid functions are considered where inputs from empirical and analytical data can provide complementary insights into housing performance. This was not done due to the lack of empirical or analytical functions developed for the area. Additionally, we have used generic structural repair cost ratios given that none has been developed yet for the Philippine context. While we posit that these are reasonable proxies, the nuances of how these ratios can vary in specific geographical contexts are worth factoring into future loss estimations once these become available. Our study dealt with housing stock distributions, but our analysis did not incorporate site exposure multipliers. This is because it is uncertain where houses will be built, and we assumed that these are household decisions that cannot be accurately predicted. This uncertainty in the spatial distribution of the houses also inhibited us from modelling the spatial correlations of wind speeds and structural fragilities

among the housing units within the building portfolios[52]. These limitations are encouraged to be explored for future work.

## Conclusion

Direct loss estimations are crucial to anticipate hazard impacts, but these are rarely conducted on a community level where localised disaster events bring a stream of considerable losses. The multi-hazard realities that people experience in these settings further complicate strategies to mitigate disaster impacts. In this study, we applied multi-objective optimisation to simulate direct economic losses of a housing stock against wind and seismic impacts based in Itbayat, Batanes in the Philippines. Monte Carlo simulations were used to explore different housing stock scenarios to identify the ideal concentration of housing typologies that would minimise losses for both hazards. Two cases were analysed concerning extreme hazard intensity thresholds. For both cases, results demonstrate that the baseline (or the current) housing stock distribution can be improved to sustain lower wind and seismic losses by achieving a combination of lightweight and reinforced concrete typologies at varying proportions. A set of Pareto optimal solutions was identified from the optimisation inquiry. These candidate solutions were then consequently analysed against the desirability, viability, and feasibility framework to situate their practicality in the field.

Our use of multi-objective optimisation at a housing stock level on a local scale opens opportunities to collectively analyse damage impacts while incorporating granular data that refines loss estimates contextualised for a specific area. This contributes to refining national-level loss estimates usually based on general parameters. Additionally, our analysis of the Pareto

optimal solutions against the socio-technical indicators instils broader conversation among stakeholders on the concept of optimality in resource-constrained settings. The trade-offs highlighted in this study went beyond the quantified losses for both hazards, but also considered the trade-offs arising from risk perceptions, local appropriateness of solutions, and availability of resources. Most often, these considerations are neglected when policymakers and technical stakeholders decide on optimal interventions. Overall, our study contributes to advancing risk reduction strategies by offering an approach to streamline and contextualise multi-hazard loss estimates.

## Methods

Our methods are divided into two broad sections. The first part describes the simulation of direct economic losses against wind and seismic impacts under different housing stock scenarios. The second part sets out the identification of the scenarios or optimal solutions that yield minimal losses considering both hazards. The flow chart of the overall methods is presented in Fig. 6. This study utilised data from previous field investigations[39,40] compliant with protocol 2022/705 issued by the Human Research Ethics Committee at the University of Sydney and the Certificate of Precondition R2-IKSP-2022-12-21 issued by the National Commission on Indigenous Peoples of the Philippines.

### Simulating direct economic losses

To simulate the direct economic losses from seismic and wind hazards, we first estimated the average construction costs of the housing typologies through quantity surveying. Next, we derived scenarios of hypothetical housing stock distributions. Finally, we quantified and aggregated the housing stock-level direct economic losses.

**Construction cost estimates.** Estimating the value of the (undamaged) housing stock was first needed before inferring direct economic loss calculations. We conducted quantity surveying for each of the six post-earthquake housing typologies surveyed in an earlier study[39] representing the construction archetypes that are now ubiquitous in the municipality of Itbayat. These typologies are lightweight, semi-concrete, and concrete typologies (see Fig. 1), and their characteristics are described in Table 1. The first author conducted the quantity surveying, having inspected the housing typologies on site which provided familiarity with the housing characteristics. The procedures of the construction cost estimates, together with the activity on-node diagrams (Supplementary Figs S1–S6) and the resulting bill of quantities (Supplementary Tables S1–S6), are presented in the Supplementary Material.

**Deriving hypothetical scenarios of housing stock distribution.** We explored different possible combinations of the six typologies to inform hypothetical housing stock distributions (hereon referred to as "scenarios") which were later assessed for their optimal performance in reducing multi-hazard losses. In deriving these scenarios, we used the partitions package in R which uses combinatorial mathematics to enumerate all the possible combinations of integers[53]. We assumed that for a given housing stock, each typology would constitute at least 5% of the overall stock distribution. The idea of representing every typology in the stock distribution across all scenarios is grounded on the observed heterogeneity of the present housing stock, implying that each of the typologies will likely be adopted or constructed by households. Additionally, due to the various socio-economic factors affecting how households build their dwellings[40], we posit that the transition to a purely homogeneous housing stock is unlikely in the future. The 5% assumption, which returned 11,628 scenarios, allowed us to have a reasonable number of stock distributions to analyse without having to deal with excessive computational efforts for the simulation in the next steps. We translated the 11,628 scenarios, initially expressed in percentages for each typology present in the housing stock, to counts of housing units. We extrapolated the percentages of each typology in the housing stock based on a total of 971 households – the

latest data on the total number of households in the municipality as obtained from the local government as of May 2024. A "baseline" scenario was identified from the 11,628 scenarios to represent the current housing stock distribution in the community based on the distribution of the residential portfolio after the 2019 earthquakes. This baseline scenario was used in comparisons with potential future distributions.

**Quantifying direct economic losses.** To quantify direct economic losses, we first simulated the losses per house ($L_h$) for each typology in every scenario using Eq. (1) based on the loss estimation methods of Lin and Wang[54] (adopted from HAZUS[55]). We carried out Monte Carlo simulations in R where the number of simulations corresponds to the quantity of houses specified per scenario. All simulations (based on a standard of 10,000 iterations) were probabilistic to account for uncertainties in the loss estimates. We therefore used probability distributions as inputs for Eq. (1) to generate a resulting probability distribution for $L_h$. This method is referred to as uncertainty propagation where the uncertainties in the inputs "propagate" into the simulation results[56].

$$L_h = C * A * l_m \tag{1}$$

where:

$L_h$ = direct economic loss per house, in Philippine peso (PHP)
$C$ = construction cost of a house per square metre, in PHP
$A$ = floor area of a house, in square metre
$l_m$ = loss multiplier

The construction cost of a house per square metre ($C$) was derived from the quantity surveying discussed in "Construction cost estimates" section. We used a triangular distribution common in probabilistic construction cost estimates which can intuitively account for a range of cost values based on the mode, the minimum, and the maximum estimates[57]. We used the estimated construction cost as the mode and ±10% for the maximum and minimum values. The 10% assumption is based on a common rule of thumb in residential project estimation in the Philippines accounting for contingencies of estimates and the inherent uncertainties with the construction market. The floor area of a house ($A$) is based on data from the Philippine Statistics Authority[58,59]. In Region 2, where Itbayat is located, the floor areas of houses approximate a lognormal distribution with most of them having 30 sqm to 49 sqm[58,59]. In our simulation, we bounded the values between 5 sqm and 200 sqm to represent realistic total floor sizes. Less than five sqm is considered too small to be habitable, while more than 200 sqm is rare within the residential portfolio of Itbayat.

The loss multiplier ($l_m$) was derived using Eq. (2). Specific hazard intensities are required in this calculation representing the triggers incurring losses in relation to housing performance of particular typologies. For this study, we described the selection of these hazard intensities in "Selecting cases for Pareto optimal solutions" section.

$$l_m = \sum_{ds2}^{ds5} (P_{dsx} * R_{dsx}) \tag{2}$$

where:

$l_m$ = loss multiplier
$ds2{:}ds5$ = damage state (ds) 2 to 5
$P_{dsx}$ = probability of a housing typology in each damage state
$R_{dsx}$ = structural repair cost ratio of a housing typology in each damage state

The probabilities of a typology reaching or exceeding a damage state ($P_{dsx}$) are based on the fragility functions derived specifically for the context of Itbayat[39]. These functions – derived through an expert-driven approach – account for wind and seismic housing performance of the housing typologies constructed after the 2019 earthquakes. (For the parameters of these fragility functions, see Hadlos et al.[39]). For the structural repair cost ratios

**[Part 1. Simulating direct economic losses]**

**Step 1:**
Generate housing stock scenarios with varying combination of percentages of housing typologies in increments of 5%. (Note: LWA, LWB, ..., & RCA are the housing typologies.)

| Scenario | LWA | LWB | SCA | SCB | RCA | RCB | |
|---|---|---|---|---|---|---|---|
| 1 | 75% | 5% | 5% | 5% | 5% | 5% | *(total = 100%)* |
| 2 | 70% | 10% | 5% | 5% | 5% | 5% | *(total = 100%)* |
| ... | | | | | | | |
| 11628 | 5% | 5% | 5% | 5% | 5% | 75% | *(total = 100%)* |

**Step 2:**
Translate percentages to actual house count (based on *n* = 971 houses).

| Scenario | LWA | LWB | SCA | SCB | RCA | RCB | |
|---|---|---|---|---|---|---|---|
| 1 | 728 | 48 | 48 | 49 | 49 | 49 | *(total = 971 houses)* |
| 2 | 680 | 97 | 48 | 48 | 49 | 49 | *(total = 971 houses)* |
| ... | | | | | | | |
| 11628 | 48 | 48 | 49 | 49 | 49 | 728 | *(total = 971 houses)* |

**Step 3:**
In **each** scenario, calculate the direct economic loss per house using probability distributions as inputs for construction cost per floor area, floor area, and loss multiplier.

direct economic loss per house ($L_h$)
=
construction cost per floor area ($C$)
x
floor area ($A$)
x
loss multiplier (of a typology for a specific hazard intensity) ($l_m$)

legend:
**h#** : house#

| Scenario | LWA | LWB | SCA | SCB | RCA | RCB | |
|---|---|---|---|---|---|---|---|
| 1 | 728 | 48 | 48 | 49 | 49 | 49 | *(total = 971 houses)* |

h1 = $L_{h1}$
h2 = $L_{h2}$
...
h728 = $L_{h728}$

h729 = $L_{h729}$
h730 = $L_{h730}$
...
h776 = $L_{h776}$

h777 = $L_{h777}$
h778 = $L_{h778}$
...
h824 = $L_{h824}$

h825 = $L_{h825}$
h826 = $L_{h826}$
...
h873 = $L_{h873}$

h874 = $L_{h874}$
h875 = $L_{h875}$
...
h922 = $L_{h922}$

h923 = $L_{h923}$
h924 = $L_{h924}$
...
h971 = $L_{h971}$

$L_{total}$ = total direct economic losses in a building stock per scenario
(for a specified hazard intensity)

**Step 4:**
Add the total direct economic losses from all houses in each scenario for a specified hazard intensity. (Note: PEIS refers to the earthquake intensity scale for the Philippines.)

**[Part 2. Optimising direct economic losses]**

**Case 1: PEIS VII & 270 km/h**

| Scenario | PEIS VII | 270 km/h |
|---|---|---|
| 1 | $L_{total}$ | $L_{total}$ |
| 2 | $L_{total}$ | $L_{total}$ |
| ... | | |
| 11628 | $L_{total}$ | $L_{total}$ |

**Case 2: PEIS VIII & 300 km/h**

| Scenario | PEIS VIII | 300 km/h |
|---|---|---|
| 1 | $L_{total}$ | $L_{total}$ |
| 2 | $L_{total}$ | $L_{total}$ |
| ... | | |
| 11628 | $L_{total}$ | $L_{total}$ |

**Step 5:**
For each case, identify the Pareto optimal solutions using preference selection from the rPref package in R.

Identify Pareto optimal solutions for each case through preference selection

Identify Pareto optimal solutions for each case through preference selection

**Step 6:**
Rank the identified Pareto optimal solutions using the R-method.

Rank the Pareto optimal solutions for each case

**Step 7:**
Assess the Pareto optimal solutions against the desirability, viability, and feasibility framework.

desirability

viability    feasibility

**Fig. 6 | Flow chart of the methods.** The first part of the methods involves simulating direct economic losses. The second part relies on the simulated losses to identify and rank Pareto optimal solutions, and analyse these solutions against the DVF (desirability, viability, and feasibility) framework.

($R_{dsx}$), none have been developed yet specific to the Philippine context which could have contextualised the loss-to-repair dynamics of the different construction archetypes within the country. Therefore, we used the default values of structural repair cost ratios from HAZUS which are used as proxies

for both wind and seismic loss estimations[55,60,61]. These are 0.02 for minor damage (damage state (ds) 2), 0.10 for moderate (ds3), 0.50 for extensive (ds4), and 1.00 for complete (ds5). It is assumed that no repair costs are required for very minor damage (ds1); thus, it was omitted from the

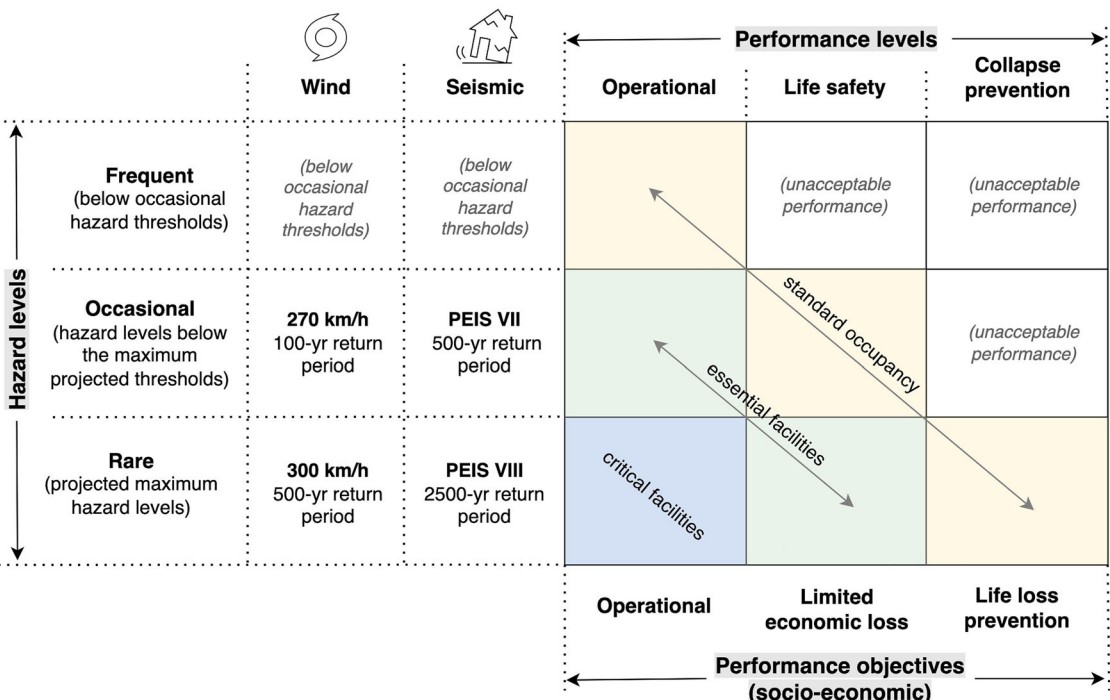

**Fig. 7 | Performance-based design matrix adapted from Elnashai and Di Sarno[62] and Tsompanakis[63].** The return periods for Itbayat, Batanes are probabilistic estimates from the Philippine Earthquake Model[64] and the Regional Severe Wind Hazard Maps of the Philippines[45]. (Note that PEIS refers to the Philippine Volcanology and Seismology Earthquake Intensity Scale).

calculation. We used the same repair cost ratios for all the typologies given that the default values are generic. A triangular distribution was adopted for the simulation where the calculated $l_m$ under the specified hazard intensity is the mode. For seismic, the minimum and maximum values are $l_m \pm$ one PEIS intensity. For wind, the minimum and maximum values are $l_m \pm 50$ kilometres per hour – an assumption based on expert judgement to establish a reasonable threshold for wind damage.

Finally, the losses per house were aggregated to quantify the total direct economic losses incurred by a housing stock in each scenario. This was calculated using Eq. (3).

$$L_{total} = (\sum_{i=1}^{n} i = L_h)_{LWA} + (\sum_{i=1}^{n} i = L_h)_{LWB} + (\sum_{i=1}^{n} i = L_h)_{SCA}$$
$$+ (\sum_{i=1}^{n} i = L_h)_{SCB} + (\sum_{i=1}^{n} i = L_h)_{RCA} + (\sum_{i=1}^{n} i = L_h)_{RCB}$$
(3)

where:

$L_{total}$ = total direct economic losses in a housing stock per scenario (for a specific hazard intensity), in PHP

$L_h$ = direct economic losses per house for a specific hazard intensity, in PHP

$n$ = number of houses for a typology in each scenario

$LWA, LWB, ... RCB$ = housing typologies

**Optimising direct economic losses**
Pareto optimality describes a situation wherein the achievement of one objective compromises compliance with the other objective/s[34]. This is a common concept in multi-objective optimisation where the inherently conflicting requirements of various objectives warrant consideration of candidate solutions that *somehow* satisfy all the required objectives. In this study, we employed Pareto optimality to identify which scenarios simultaneously yield the least direct economic losses, considering a tandem of predefined seismic and wind intensities. Building a dwelling to resist both seismic and wind impacts is an example of a multi-objective problem

requiring Pareto optimal solutions due to the competing impacts of these hazards. In the following sections, we describe how we selected hazard intensities for the Pareto optimal analysis, followed by how we identified and ranked the Pareto optimal solutions. Lastly, we present how we incorporated a framework to analyse these solutions against socio-technical indicators.

**Selecting cases for Pareto optimal solutions.** The calculation of direct economic losses is based on specific hazard intensities as earlier shown in Eq. (2). In this study, we chose two (2) pairs of wind and seismic intensities (hereon referred to as "cases") where Pareto optimal solutions were analysed. Each case represents the objectives of the inquiry in that we aim to simultaneously minimise the direct economic losses incurred by a housing stock respective to the specified intensities. The selection of the pairs of intensities does not imply the simultaneous co-occurrence of these. Rather, we selected the cases as thresholds where losses to housing assets are anticipated, guided by the performance-based design matrix[62,63] (see Fig. 7). While originally conceptualised for seismic applications, we extended the use of the performance-based design matrix to wind analysis. Given that the concept of the matrix generically establishes and maps (expected) performance levels of building/occupancy types against probability of hazard occurrences, it can then be adapted for other hazard contexts beyond earthquakes. For example, under "occasional" hazard intensities as described in Fig. 7, a basic occupancy building (e.g., a residential dwelling) is recommended to meet life safety (or damage control) requirements – a consideration we deem is also applicable to other hazards, not just to seismic. Building codes provide regulations linking specific probabilistic hazard frequencies to required performance levels (e.g., life safety requirement for seismic hazards having 475-year return period). However, with emerging hazard projections coupled by the lived experiences of communities in disaster-affected regions, prescribed thresholds of hazard frequencies from building codes can be adapted in the performance-based design analysis to reflect both analytical and empirical sources of information.

The cases were based on hazard levels, with the pairing of intensities drawn from the corresponding return periods. Our analysis is geared

towards life safety and collapse prevention – the performance level requirements for residential dwellings (standard occupancy) when these are subjected to occasional and rare hazard intensities. The return periods are probabilistic estimates from the Philippine Earthquake Model[64] and the Regional Severe Wind Hazard Maps of the Philippines[45]. Seismic intensities are based on the Philippine Volcanology and Seismology Earthquake Intensity Scale (PEIS) which is the nationally developed earthquake intensity measurement in the Philippines[41]. Meanwhile, wind intensities are expressed in kilometres per hour (km/h) and represent a 3-second peak gust measured 10 m above ground[45].

The first case is PEIS VII and 270 km/h representing occasional hazard intensities. "Occasional" in this context implies hazard levels below the threshold of the most extreme hazard intensities projected. A 270 km/h wind speed has a probability of 1% (100-year return period) to 5% (20-year return period) of occurring in a given year[45]. Meanwhile, a peak ground acceleration of 0.3–0.4 g (approximately PEIS VII[42–44])) has a probability of 0.2% occurring annually (500-year return period)[64]. These intensities align with high-impact disaster events experienced in the community. In 2016, Typhoon Ferdie (Meranti) hit mainland Batanes with a gustiness of up to 252 km/h, then made landfall in Itbayat where its intensity (was believed to) peaked but no official records are available due to limitations of weather instruments[65] (potentially reaching ~270 km/h). In 2019, a series of earthquakes struck the island, and the majority of the residential areas were exposed to the impacts of PEIS VII[37,38]. Thus, these intensities can also be referred to as the experiential maximums – a threshold that will likely inform policies of disaster mitigation based on the first-hand encounters of local constituents.

The second case is PEIS VIII and 300 km/h representing rare hazard intensities. "Rare" in this context implies the anticipated maximum hazard levels in the probabilistic hazard maps. The projected maximum wind speed that can be experienced in Itbayat is 300 km/h simulated to have a 0.2% likelihood to occur in any given year (500-year return period)[45]. Meanwhile, the maximum peak ground acceleration expected is 0.6 g (approximately PEIS VIII[42–44])) which has a 0.04% likelihood in a given year (2500-year return period)[64]. No official records can confirm if these wind and seismic intensities have been experienced historically in Itbayat. These projections of intensities – being the theoretical maximums in our analysis – are useful to anticipate the vulnerability of structural assets of the residential building portfolio under extraordinary circumstances.

**Identifying and ranking Pareto optimal solutions.** The Pareto optimal solutions were identified for each case by determining the scenarios that simultaneously yield the least direct economic losses for both wind and seismic hazards. To do this, we used the preference selection (*psel*) function in rPref package in R[66]. The *psel* function evaluates a preference from a given data set. In this study, the preference was to identify the scenarios with the least losses considering both hazards, returning a set of non-dominated solutions (or Pareto optimal solutions) that satisfy this preference. The process of deriving these non-dominated solutions involves an algorithm that iteratively evaluates all possible outcome combinations and returns the best solutions being those that are not dominated by any other possible solutions. The loss outputs from Eq. (3) are probability distributions, thus the 95th percentile values for these distributions were selected for the preference selection analysis. We used the 95th percentile value being more stringent and conservative considering the high-risk consequences that could possibly arise from underreporting potential worst-case outcomes in probabilistic loss analysis.

The Pareto optimal solutions identified for each case were then ranked to filter the top solutions. We used the multi-attribute decision-making method proposed by Rao and Lakshmi[46] called the R-method. This method has been demonstrated to outperform widely used multi-attribute decision-making tools such as the simple additive weighting (SAW), weighted product method (WPM), among others[46]. The first step in using the R-method is to rank the objectives according to perceived importance. Here, we assigned a 1.5 rank (average of 1st and 2nd ranks) for both the wind and seismic objectives implying their equal importance because impacts from both hazards do bring considerable damage to structural assets. Next, the Pareto optimal solutions were ranked for each hazard. Since the goal is to minimise losses, the solution with the lowest direct economic loss was ranked 1st and the highest was ranked last. The ranks for both objectives and solutions were then converted to their corresponding weights using Eq. (4). Examples of deriving these weights can be found in Rao and Lakshmi[46].

$$w_j = \frac{\left( \frac{1}{\sum_{k=1}^{j} \left( \frac{1}{r_k} \right)} \right)}{\sum_{j=1}^{n} \left( \frac{1}{\sum_{k=1}^{j} \left( \frac{1}{r_k} \right)} \right)} \qquad (4)$$

where:

$w_j$ = weight of objective/solution $j$ ($j$ = 1, 2, 3, …, n)
$r_k$ = rank of objective/solution $k$ ($k$ = 1, 2, 3, …, j)
$n$ = number of objectives/solutions

The weights of the solutions were then multiplied by the weight of their corresponding objective. The products of these were summed up across each scenario to create a composite score where the score with the highest value/s represent/s the top solution/s.

**Contextualising the Pareto optimal solutions against the DVF framework.** The Pareto optimal solutions represent the candidate scenarios that best address the objectives of simultaneously minimising direct economic losses against wind and seismic impacts. However, these solutions might not be pragmatically applicable without analysing them against the socio-technical factors affecting the residential construction context. Hence, we contextualised the Pareto optimal solutions against the desirability, viability, and feasibility (DVF) framework. The DVF framework is used in design thinking and business disciplines to evaluate whether proposed innovations will be desirable to users, viable for organisations, and technically and financially feasible[67]. We adapted this framework for this study to evaluate whether the characteristics of the housing stock distributions of the Pareto optimal solutions align with the households' preferred mode of construction ("desirability"), suitable for the community in the long term ("viability"), and realistically achievable ("feasibility") (see Fig. 8). Our assessment was qualitative with the aim to generally appraise potential solutions within practical considerations. This qualitative assessment relied on primary data from an earlier study in the same field study site which used field immersion, interviews, and focus group discussions to understand the socio-technical factors of housing reconstruction within the selected community[40]. Note that desirability, viability, and feasibility are dynamic lenses influenced by circumstances, such that households' perceptions can change or new technologies or increased financial capacities can shift what is viable and/or feasible.

We posit that concrete structures are favoured most by households in Itbayat based on a previous study where households increasingly seek to construct heavier, reinforced typologies following the seismic events in 2019 and the noticeably intensifying wind hazards[40]. Restrictions on cutting and sourcing hardwood grown on the island have also forced households to use commercially available alternatives which raised scepticism about whether alternative timber materials are as durable as the local species of wood they commonly used before. Hence, we assessed scenarios as desirable if most houses are concrete-based typologies (SCA, SCB, RCA, RCB). Desirability is a critical aspect as previous studies in different contexts have shown how households' perception of housing safety greatly influences the adoption of (and receptiveness to) certain construction archetypes[48,68]. Factoring in desirability also eliminates the risk of endorsing (and eventually imposing) unwanted solutions common in the developmental context.

**Fig. 8 | The desirability, viability, and feasibility (DVF) framework.** The desirability criteria is based on households' preferred mode of construction. The viability criteria pertains to longevity of construction materials associated with building a certain housing typology. The feasibility criteria considers the technical and financial aspects of house construction.

**Desirability**

Do the households prefer constructing majority of the typologies leading to more receptive adoption?

**Viability**

Are most of the typologies durable enough to thrive in the local geographical context?

**Feasibility**

Can most of the typologies be constructed given the available financial and technical resources/circumstances?

Ideal scenario

We assessed viability based on the idea of permanence associated with lesser maintenance requirements and inherent longevity of typologies. Lightweight typologies tend to incur maintenance more frequently than concrete structures and can be less durable when exposed to extreme environmental conditions[69]. For example, corrugated galvanised iron sheets used as the main wall cladding for lightweight houses can eventually be prone to rusting due to sea breeze (Itbayat being an island) on top of their susceptibility to physical damage. Additionally, (untreated) wooden materials can easily become structurally compromised when exposed to damp conditions. We thus have considered scenarios with less lightweight houses to be more viable.

Feasibility was assessed on economic and technical bases. Cost-wise, the heavier the typologies become, the costlier they are to construct. On average, the differential cost of building a fully lightweight and a fully concrete house in Itbayat is ~ PHP 600,000 (USD 10,700) – a cost not commensurate to be easily funded (without formal subsidies) when most of the households rely on fishing and farming for their livelihoods. The high construction costs on the island have been primarily due to the logistical challenges of importing materials and the regulatory barriers affecting the extraction of locally available materials forcing locals to rely on costly commercial alternatives[40]. On a technical basis, local building expertise in Itbayat has historically been focused on wooden construction systems, with the majority of the available construction labour on the island more well-versed in timber construction. Given these considerations, we assessed a scenario to be more feasible if fewer houses are of full or partial concrete construction.

## Data availability

All data required to reproduce the findings are presented in the manuscript and in the supplementary file.

## Code availability

The code used in this study is available at https://github.com/arvinhadlos/loss-estimation.git.

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

## Author contributions

A.H. wrote the main manuscript text, conducted the data collection, data analysis, and data visualisation. A.O. provided primary supervision while S.A.H. provided secondary supervision. A.H. and A.O conceptualised the study. All authors reviewed and edited the manuscript.

## Competing interests

The authors declare no competing interests.
