## [Transparent Peer Review file · Communications Engineering]

Optimising housing typology distributions for multi-hazard loss reductions in resource-constrained settings

Corresponding Author: Mr Arvin Hadlos

Version 0:

Reviewer comments:

Reviewer #3

(Remarks to the Author)

The manuscript presents an interesting approach to assess "optimal" combination of earthquake (EQ) and tropical cyclone (TC) typologies with the aim of reducing disaster risk. However, it does not make any consideration of the costs associated to such combinations and therefore, despite its good intentions, a key ingredient is missing. My recommendation is to reject the study in its current state but I strongly encourage the authors to address, even with an approximate method, the cost dimension. Below, I'm providing some other points authors may find useful:

- The motivation of the study is solid even if its applicability because lacking a consideration of the costs is limited.
- What rare and occasional hazard occurrences should be described in terms of return periods in the body of the manuscript.
- The EQ intensity measure chosen, does not have a good correlation with the losses, if compared for instance with spectral accelerations. This must be at least mentioned.
- Authors should note, however, that hazard intensities associated to a return period do not correspond to a single event but to the contribution of many; which is particularly true in the case of the Phillipines. Therefore, the frequency/occurrence component of the approach needs to be revised.
- Within the loss assessment framework adopted (after HAZUS), it is not clear how the spatial correlation for it is being considered (if at all).

Reviewer #4

(Remarks to the Author)

Dear authors, please find my comments in the attached document. Best wishes, Gemma Cremen
(This referee report is appended to the end of the transparent peer review file)

Version 1:

Reviewer comments:

Reviewer #4

(Remarks to the Author)

I thank the authors for comprehensively responding to my original comments. I only have a few outstanding minor comments for the authors to address, which are all based on their responses to the first review:

1. Response to Comment 1a:

The authors provide a strong justification for the selection of hazards considered in their rebuttal letter, mentioning in-person consultations and the municipality's land use plan; this justification should also be incorporated into the text itself

2. Response to Comment 1b:

The authors should provide justification for how the performance-based design matrix is relevant to this context, especially given that the matrix was originally designed for earthquake hazards exclusively

Version 2:

Reviewer comments:

Reviewer #4

(Remarks to the Author)

I thank the authors for responding to my remaining comments. Overall, I am satisfied with the rebuttal. However, I would think that the justification for extending the application of the performance-based design matrix (in response to my second comment) could be slightly better weaved into the existing text. As it is, the justification mentions the concept of an occasional hazard intensity two paragraphs before it is explained; in my view, readability of the manuscript would be improved if the explanation was placed in the same paragraph. Perhaps the authors could first dedicate a paragraph to explaining and justifying the performance-based design matrix, including a description of the hazard intensity levels used. Then, the subsequent paragraphs could be dedicated to explaining and justifying the specific values of the hazard intensities used for each hazard in the case study.

Responses to Reviewer Comments

We would like to thank the reviewers for the thoughtful comments and constructive suggestions which were valuable in improving the quality of this manuscript. We have carefully addressed each comment below. We list the review comments verbatim in bold, followed by our responses to these comments in plain text. We italicise with quotations key additions to the manuscript.

REVIEWER #3

The manuscript presents an interesting approach to assess "optimal" combination of earthquake (EQ) and tropical cyclone (TC) typologies with the aim of reducing disaster risk. However, it does not make any consideration of the costs associated to such combinations and therefore, despite its good intentions, a key ingredient is missing. My recommendation is to reject the study in its current state but I strongly encourage the authors to address, even with an approximate method, the cost dimension.

We welcome the reviewer's comment on the cost dimension of the housing stock transitions. We reflect on a few important points which relate to the cost before introducing further discussion in the manuscript on this topic.

- (1) The cost dimension of housing stock transitions forms part of the DVF framework in that we acknowledge the financial feasibility parameter to make transitions realistic alongside the economic circumstances of the households (see Methods > Optimising direct losses > Contextualising Pareto optimal solutions). However, we took a qualitative and descriptive approach with the general aim to contextualise plausible housing stock transitions. In the next two points below, we give grounds for the exclusion of the exact costs associated with housing stock transitions, justifying our adoption of a qualitative descriptor of economic aspects rather than calculating the particular costs associated with intended transitions.
- (2) We interpret scenarios of transitions based on *ratios* of housing typologies (and not on actual house counts) given that we envision that transitions can take place amid the dynamic need to construct more houses for endogenous reasons (i.e. due to potential shifts in population, household sizes, living preferences, etc.). These ratios are meant as a guideline to regulate (enforce/inhibit) the construction of specific typologies. For example, if the target is to transition to 75% LWA from the baseline of 20% LWA, building officials can encourage more construction of LWA, limit the construction of other typologies, or actively seek to retrofit. We refer this as typology allocations on an as-needed, continuing basis – a strategy that can be implemented over time through development controls such as permitting processes or more active incentive programs (i.e. training, rebates, etc). Thus, quantifying a fixed cost of a housing stock for transitions in dynamic – and aptly, "incremental" context – obscures the inherent idea central to this study that these changes happen longitudinally, affected by the flow and access of resources in the community over time (see Discussion section). To clarify, we have initially calculated multi-hazard losses based on actual house counts to determine the synergies of actual losses between wind and seismic hazards, but the resultant housing stock combinations are then interpreted as ratios to account for the dynamic transitions over time as explained above.
- (3) The presence of a baseline scenario in our analysis presupposes that a transition does not happen on a "blank slate" – meaning that households need to decide housing upgrades relevant to what housing typology they currently have (unless an entirely new construction is warranted, finances considered). Thus, the transition is expected to be – cost-wise – not straightforward given two possible instances, especially pronounced in resource-constrained settings. First, closely classed typologies can be upgraded from/to each other by modifying some components while retaining (or reusing the same materials for) other elements. For example, LWA can be upgraded to LWB by replacing timber posts with steel posts, all while retaining/reusing roofing and building envelopes. Second, differently classed typologies can be upgraded with an entirely or partially new set of building materials. For example, LWA to RCB transition requires almost entirely new materials while SCB to RCA can be done incrementally, retaining/reusing portions

of the existing construction. These two instances highlight the complexity of quantifying the cost of housing transition in resource-constrained settings given the heterogeneity of the housing stock, and the inherent nuances associated with construction expenses between housing stock scenarios. More realistically, the specific pathways of housing transition are personal household decisions which are difficult to predict and we believe beyond the scope of this study. Had the circumstance been a blank slate in a formal construction market (e.g., donor-provided housing relocation in a greenfield site), deriving and comparing the construction costs between/among ideal scenarios would be much more straightforward. To this case, the bill of quantities for the construction cost per floor area for each typology as provided in the Supplementary File would be helpful to have an approximation of the housing stock values.

We understand that we might not have explained clearly the points discussed above. Therefore, we have updated the following sections in the manuscript.

Discussion section, paragraph 3 – *“The housing stock transitions proposed in this study can be implemented through the variable increase and decrease of housing typology quantities (based on the prescribed ratio of housing stock combinations). This is a longitudinal process due to the dynamic housing needs influenced by the (expected) demographic changes over time, such as shifts in population, household sizes, and living setups or preferences. As such, this is a sustained effort that can be championed by planning and engineering officials to work towards optimal stock distributions within their jurisdictions as a form of disaster mitigation. Increasing certain typology quantities means encouraging new construction or incremental housing modifications towards desired typologies, all while controlling or inhibiting the construction of other typologies (thus “decreasing” their ratios relative to the uptake of the new construction or housing modifications). In practice, a starting point for this to happen is through having community-wide construction guidelines enforceable through development approvals and permitting processes. It must be noted that housing stock transitions in resource-constrained settings are not straightforward processes [47] because construction assets might not be accrued easily due to socio-demographic circumstances. Below, we outline some practical strategies for housing transitions to take place realistically.”*

Below, I'm providing some other points authors may find useful:

- The motivation of the study is solid even if its applicability because lacking a consideration of the costs is limited.

We thank the reviewer for acknowledging the robustness of the motivation behind this study. As for the cost dimension, please see our response above.

- What rare and occasional hazard occurrences should be described in terms of return periods in the body of the manuscript.

While we have expounded on what we mean by “rare” and “occasional” hazard occurrences (including their corresponding return periods) in the Methods section, we acknowledge that these are not explained in the main body of the manuscript. We have, therefore, explained these terms in the Introduction section to operationalise the terms at the start of the manuscript.

Introduction section > paragraph 8 – *“[...] The first case of our analysis represents occasional hazard occurrences (PEIS VII and 270 km/h) pertaining to hazard levels below the maximum projected thresholds. The second case is rare hazard occurrences (PEIS VIII and 300 km/h), which represents the projected maximum hazard intensities. [...]”*

- The EQ intensity measure chosen, does not have a good correlation with the losses, if compared for instance with spectral accelerations. This must be at least mentioned.

The nature of the only available fragility functions for Itbayat (the case study site) limited us to use the macroseismic scale for loss estimations. In rural Philippine contexts (e.g., in Itbayat), spectral accelerations alone are rarely used to infer the relationship between hazard intensities and corresponding losses due to technical and resource limitations of adopting spectral accelerations. We

believe that the use of macroseismic intensities to analyse losses is reasonable not only because of the study context considered but also because of the recent advances in understanding the loss relationships between seismic intensities and ground accelerations. For example, loss correlations between macroseismic intensities and ground accelerations have been discussed for loss modelling by the Global Earthquake Model (GEM) with the publication of the “Best Practices for Using Macroseismic Intensity and Ground Motion Intensity Conversion Equations for Hazard and Loss Models”. Additionally, in the Philippines, macroseismic intensity readings are already based on a country-wide “earthquake intensity meter network” which relies on “intensity meters” to determine impacts of ground shaking based on the combination of inputs of felt intensities and peak ground accelerations (see [44]). To reflect the relationships of macroseismic intensity and spectral accelerations, we have included in our revised manuscript the approximate equivalents of PEIS to peak ground accelerations.

Introduction section > paragraph 8 – “[...] PEIS range from Intensity I (scarcely perceptible; approximate peak ground acceleration (PGA) of < 0.0005 g; equivalent to Modified Mercalli Intensity (MMI) I) to Intensity X (completely devastating; approximate PGA of > 1.39 g; equivalent to MMI XII) (see [41], [42], [43], [44]). [...]”

- Authors should note, however, that hazard intensities associated to a return period do not correspond to a single event but to the contribution of many; which is particularly true in the case of the Philippines. Therefore, the frequency/occurrence component of the approach needs to be revised.

We apologise if any text implied that return periods correspond to a single event as this was not intended. We would like to clarify our selection of hazard intensities in our analysis. We have selected the tandem of hazard intensities based on the performance-based design matrix which we now have visualised for clarity. We conceptualised “rare” hazard intensities to be the projected maximum hazard levels based on probabilistic wind and seismic maps in the Philippines. Meanwhile, we considered “occasional” to be those hazard levels below the maximum projected thresholds. In respect to building codes, our analysis was geared towards life safety and collapse prevention – the performance level requirements for residential dwellings (categorised under standard occupancy) when these are subjected to occasional and rare hazard intensities. The idea is that under both rare and occasional hazard levels, we aim for the socio-economic performance objectives of limited economic loss and life loss prevention. We would like to emphasise that, as mentioned in the original manuscript, the selection of paired wind-seismic intensities is based on the thresholds where anticipated losses to housing assets are expected, and not on the possibility of the temporal, simultaneous co-occurrence of these hazard intensities. Thus, we have paired what could be considered rare and occasional for both hazards.

Methods section > Optimising direct economic losses > Selecting cases for Pareto optimal solutions – “[...] The selection of the pairs of intensities does not imply the simultaneous co-occurrence of these hazards. Rather, we selected the cases as thresholds where losses to housing assets are anticipated, guided by the performance-based design matrix [62], [63] (see Figure 7). The cases were based on hazard levels, with the pairing of intensities drawn from the corresponding return periods. Our analysis is geared towards life safety and collapse prevention – the performance level requirements for residential dwellings (standard occupancy) when these are subjected to occasional and rare hazard intensities. [...]”

Figure 7. Performance-based design matrix adapted from Elnashai & Di Sarno [62] and Tsompanakis [63]. The return periods for Itbayat, Batanes are probabilistic estimates from the Philippine Earthquake Model [64] and the Regional Severe Wind Hazard Maps of the Philippines [45].

- Within the loss assessment framework adopted (after HAZUS), it is not clear how the spatial correlation for it is being considered (if at all).

We have mentioned in the original manuscript (see Discussion section > last paragraph) that one of the limitations of our study is the exclusion of spatial correlations given that we are dealing with hypothetical housing stock distributions, and thus, there is uncertainty as to where these structures will be built specifically. To quote, “Our study dealt with housing stock distributions, but our analysis did not incorporate site exposure multipliers. This is because it is uncertain where houses will be built, and we assumed that these are household decisions that cannot be accurately predicted. This uncertainty in the spatial distribution of the houses also inhibited us from modelling the spatial correlations of wind speeds and structural fragilities among the housing units within the building portfolios (see [52]). These limitations are encouraged to be explored for future work.”

REVIEWER #4

This study examines the multi-hazard risk associated with different configurations of housing types in a municipality of the Philippines. The authors determine configurations that would lead to lower multi-hazard losses than the current housing stock distribution. Furthermore, the best configurations are further analysed against a desirability, viability, and feasibility DVF framework to shed light on whether they could be implemented in practice. In summary, I believe this is an interesting contribution to the disaster engineering risk literature that would appeal to readers of this journal. I particularly welcome the authors’ use of the DVF framework, which provides the study with some practical grounding. Having said this, I believe that changes to the manuscript are necessary before it could be deemed publishable, as detailed in the comments below.

We thank the reviewer for their constructive comments which helped us improve the quality of the manuscript. Below, we have addressed the concerns raised by the reviewer.

Main comments:

1. One of my biggest concerns centers on the hazard component of the analysis, which needs to be substantially further justified and elaborated.

a. Firstly, the authors need to justify their focus on wind and earthquake hazard. What about other hazards that the Philippines is prone to, like floods and landslides? If this approach were to be replicated in other contexts, how should hazards be selected for consideration?

Wind and earthquake hazards are the only known hazards to have direct impacts on housing assets in the municipality of Itbayat, Batanes as stated in the Comprehensive Land Use Plan of Itbayat and further corroborated by in-person consultations with the Municipal Engineering Office and the Municipal Planning and Development Office. Droughts happen but have more direct impacts on crops and livelihoods, rather than to housing assets. Flooding and storm surge are not a concern given the cliffside topography of the island municipality. Landslides are not an imminent threat given the limestone geology of the area (Itbayat was formed via coral uplift). We have provided a concise statement to reflect this consideration of focusing on wind and seismic hazards.

If our approach were to be replicated in other contexts where there are more than two prominent hazards that inflict damage to housing assets, multi-objective optimisation can account for these additional hazards by adding the desired variables. The approach would be the same, and with more than four hazards (or “objectives”) considered, this will now be called “many-objective optimisation”. Our work presents an approach to analyse multi-hazard losses regardless of the number of hazards considered.

Introduction section > paragraph 7 – “[...] We focused on wind and seismic hazards because they are the most prominent hazards and the only known hazards that inflict significant damage to housing assets in the selected municipality.”

b. I urge the authors to reflect and further elaborate on the selection of specific hazard scenarios. How were the selected return periods deemed to be suitably representative of “occasional” versus “rare” classifications, why are just two return period pairs considered sufficient for this analysis (why not use a fully probabilistic hazard approach, for instance), how do they relate to any building codes implemented, and how are 0.2% and 1-5% probabilities of occurrence (in the case of the occasional classification) deemed to be compatible? In my opinion, the entire analysis lacks a strong rationale unless the previous questions can be clearly answered by the authors.

We have selected the tandem of hazard intensities based on the performance-based design matrix which we now have visualised for clarity. We conceptualised “rare” hazard intensities to be the projected maximum hazard levels based on probabilistic wind and seismic maps in the Philippines. Meanwhile, we considered “occasional” to be those hazard levels below the maximum projected thresholds. In respect to building codes, our analysis was geared towards life safety and collapse prevention – the performance level requirements for residential dwellings (categorised under standard occupancy) when these are subjected to occasional and rare hazard intensities. The idea is that under both rare and occasional hazard levels, we aim for the socio-economic performance objectives of limited economic loss and life loss prevention. We would like to emphasise that, as mentioned in the original manuscript, the selection of paired wind-seismic intensities is based on the thresholds where anticipated losses to housing assets are expected, and not on the possibility of the temporal, simultaneous co-occurrence of these hazard intensities. Thus, we have paired what could be considered rare and occasional for both hazards.

Methods section > Optimising direct economic losses > Selecting cases for Pareto optimal solutions – “[...] The selection of the pairs of intensities does not imply the simultaneous co-occurrence of these hazards. Rather, we selected the cases as thresholds where losses to housing assets are anticipated, guided by the performance-based design matrix [62], [63] (see Figure 7). The cases were based on hazard levels, with the pairing of intensities drawn from the corresponding return periods. Our analysis

is geared towards life safety and collapse prevention – the performance level requirements for residential dwellings (standard occupancy) when these are subjected to occasional and rare hazard intensities. [...]"

Figure 7. Performance-based design matrix adapted from Elnashai & Di Sarno [62] and Tsompanakis [63]. The return periods for Itbayat, Batanes are probabilistic estimates from the Philippine Earthquake Model [64] and the Regional Severe Wind Hazard Maps of the Philippines [45].

2. The results of the analysis are highly dependent on the fragility functions used. However, very little (if any) detail on these functions is provided by the authors. Please explain (for both hazards) (1) the applicability of the functions used for various typologies (including details on the data from which they are derived). This is especially important given the “heterogeneity of the housing stock” and “plurality of how households constructed their dwellings”; (2) the parameters of the functions used for each typology; and (3) the damage states captured in the fragility functions. Without these details, a judgement on the reliability of the study’s outcomes cannot be made.

In the Methods section, we have mentioned and cited that the fragility functions used were based on our earlier study in Itbayat (the same case study site) focusing on deriving fragility functions specific to this context (see: <https://doi.org/10.1038/s41598-023-49398-2>). All the necessary information (e.g., beta values of the curves, damage state descriptions, etc.) is contained in this publication. Given the bulk of such information, we believe that citing this publication is an appropriate way to refer to the necessary details regarding the fragility functions used. We have, however, still added a brief statement in this manuscript to highlight the past work we cited.

Methods section > Simulating direct economic losses > Quantifying direct economic losses – “[...] The probabilities of a typology reaching or exceeding a damage state (P_{dsx}) were based on the fragility functions derived specifically for the context of Itbayat (see [39]). These functions – derived through an expert-driven approach – account for wind and seismic housing performance of the housing typologies constructed after the 2019 earthquakes. (For the parameters of these fragility functions, see [39].)”

3. The discussion section requires some improvement, in my opinion:

a. The absence of hazard multipliers is not the only limitation of the hazard analysis; please revert to my first main comment

We have addressed the first main comment above. As for the other limitations of our study (apart from the absence of hazard multipliers), we have briefly outlined them as follows: (i) use of expert-driven fragility functions which could still be further substantiated with analytical and empirical functions; (ii) reliance on generic structural repair cost ratios (from HAZUS) due to the non-availability of repair cost ratios tailored for the Philippine context; and (iii) the exclusion of spatial correlations of wind and seismic fragilities due to the limitations in inferring the locations of the houses. These limitations were outlined in our original manuscript (see Discussion > last paragraph).

b. Line 421 page 11: don't these statements contradict the whole point of this study – i.e., the fact that we should not be considering hazards in isolation from each other?

We have revised this statement to be consistent with the goal of the study to not consider hazard impacts in isolation.

Discussion section > paragraph 7 – “[...] For example, no-build zones can be enforced where wind exposure is the strongest and seismic codes can be reviewed where they benefit soil-structure relationships while considering wind-seismic trade-offs accordingly towards decision-making. [...]”

c. The concept of resilience enhancement arises here (as well as in other sections of the text). However, enhancing resilience of the building stock requires much more than exclusive consideration of immediate economic losses – recovery time and recovery trajectories are significant components of resilience that are not dealt with in this study. Therefore, I do not think that the authors should be emphasizing the contribution of this study to resilience enhancement.

We have carefully reviewed excerpts where the word “resilience” is mentioned, and we have accordingly revised these to “risk reduction” (or similar).

Introduction section > paragraph 3 – “[...] Due to the differing dynamics of counteracting multiple hazards, investments towards structural risk reduction against only one hazard can create risk to other hazards.”

Introduction section > paragraph 4 – “[...] This is critical in multi-hazard settings where structural risks are being complicated by the exposure of assets to independently occurring hazards having different dynamics, frequencies, and impacts, obscuring straightforward pathways towards risk reduction.”

Introduction section > paragraph 5 – “[...] This larger unit of analysis is poised to support community efforts towards risk reduction through the potential regulation of structural typologies in heterogeneous building stocks susceptible to multi-hazard exposure.”

Introduction section > paragraph 7 – “The main objective of this study is to explore transitions towards more optimal housing stock distributions, which simultaneously minimise direct economic losses from both wind and seismic hazards. [...]”

Results section > Case 1 > paragraph 3 – “[...] While the closest optimal solution to the baseline might not be desirable given that it is predominantly lightweight, this scenario is the most practical trajectory to lessen multi-hazard losses.”

d. I acknowledge the authors' discussion on how the transition of the housing stock can be achieved. However, I struggle to understand how the “ideal ratio” would work in practice. Which households would be incentivized to achieve which typology, and how would the “ideal ratio” be consistently enforced?

We have added a paragraph to briefly explain how the ideal ratios would work in practice and how these ratios can be enforced. To clarify, “incentives” are meant to be a generic approach to encourage construction or housing modifications towards the desired typologies – thus, it is not intended to pinpoint which “households” qualify to get an incentive. In our original manuscript, we have provided suggestions on how transitions would take place to provide plausible typology-to-typology transitions independent of whether these will be incentivised or not.

Discussion section > paragraph 3 – “The housing stock transitions proposed in this study can be implemented through the variable increase and decrease of housing typology quantities (based on the prescribed ratio of housing stock combinations). This is a longitudinal process due to the dynamic housing needs influenced by the (expected) demographic changes over time, such as shifts in population, household sizes, and living setups or preferences. As such, this is a sustained effort that can be championed by planning and engineering officials to work towards optimal stock distributions within their jurisdictions as a form of disaster mitigation. Increasing certain typology quantities means encouraging new construction or incremental housing modifications towards desired typologies, all while controlling or inhibiting the construction of other typologies (thus “decreasing” their ratios relative to the uptake of the new construction or housing modifications). In practice, a starting point for this to happen is through having community-wide construction guidelines enforceable through development approvals and permitting processes. It must be noted that housing stock transitions in resource-constrained settings are not straightforward processes [47] because construction assets might not be accrued easily due to socio-demographic circumstances. Below, we outline some practical strategies for housing transitions to take place realistically.”

e. The discussion on limitations ought to mention that the DVF framework has been calibrated based on secondary data

The DVF framework is based on an earlier study we conducted in the same case study site (<https://doi.org/10.1016/j.jobe.2024.109636>). This was a field-based study done ethnographically to elicit the housing reconstruction pathways among the Indigenous households. As such, we would like to clarify that the DVF framework is not based on a secondary data.

Methods section > Contextualising Pareto optimal solution – “[...] Our assessment was qualitative with the aim to generally appraise potential solutions within practical considerations. This qualitative assessment was based on primary data from an earlier study in the same field study site which used field immersion, interviews, and focus group discussions to understand the socio-technical factors of housing reconstruction within the selected community (see [40]). [...]”

Technical comments on the methodology:

1. Why 11,628 scenarios? (I assume this has something to do with the 5% increments used, but please specify). Update: I see this is now clarified in Line 515, page 14 but I think the clarification should come earlier, where the 11,628 value is first introduced. I think you should also make it clear in the main text (on page 3) that an explanation for the number of scenarios considered is provided in the methods section.

We thank the reviewer for this suggestion as this helps to position early on why we analysed 11,628 scenarios.

Introduction section > paragraph 8 – “[...] We generated 11,628 housing stock scenarios based on the assumption that for a given housing stock, each typology would constitute at least 5% of the overall stock distribution. We then simulated the loss outputs for each scenario using Monte Carlo simulation, considering two cases of paired extreme hazard intensity thresholds. [...]”

Introduction section > paragraph 8 – “[...] For the entire data analysis procedures of this study, see “Methods” section.”

2. Line 511-513, page 14: “This assumption is grounded on the observed heterogeneity of the present housing stock, implying that each of the typologies will likely be adopted or constructed by communities.” It is not clear how this statement is connected to the 5% assumption made.

We used 5% increments to reduce computational requirements while allowing us a reasonable number of stock distributions to analyse. The assumption on the heterogeneity is linked to the idea that each typology would constitute at least 5% of the stock distribution, considering that 5% is the least ratio assumed for computational purposes. This heterogeneity therefore implies that all typologies are accounted for in every scenario of housing stock distributions. To address this confusion, we have improved the wording for this particular statement.

Methods > Simulating direct economic losses > Deriving hypothetical scenarios... – “[...] We assumed that for a given housing stock, each typology would constitute at least 5% of the overall stock distribution. The idea of representing every typology in the stock distribution across all scenarios is grounded on the observed heterogeneity of the present housing stock, implying that each of the typologies will likely be adopted or constructed by households.”

3. Line 666: Why use the 95th percentile values – why not use the expected loss values, for instance? It would be useful to include a sensitivity analysis to understand if the results change depending on the loss statistic used.

Our loss outputs are probabilistic to account for the uncertainty in our loss estimates. Thus, as initially provided in our manuscript, we presented the histograms of the loss estimates. By presenting the probabilistic loss values, decision-makers can opt to look at particular values (out of the range of losses presented) vis-à-vis their risk appetite. We have used the 95th percentile as a conservative and stringent value given that underestimating anticipated losses in high-risk settings presents high consequences that could be mitigated by regarding potential worst-case outcomes as part of decision-making strategies. We reflected this justification in the revised manuscript.

Methods section > Optimising direct economic losses > Identifying and ranking Pareto optimal solutions – “[...] We used the 95th percentile value considering the high-risk consequences that could arise from underreporting potential worst-case outcomes in probabilistic loss analysis.”

More minor comments:

1. The title of the manuscript seems too obvious, i.e., if something is optimized (on the basis of losses), it is certain to lead to reduced losses. Furthermore, it is not clear why the analysis would be specifically restricted to “resource-constrained” communities.

We have (slightly) revised our title to be more process-oriented (rather than results-based) retaining three key aspects: the method (“optimising housing typology distributions”), the objective (“multi-hazard loss reduction”), and the context (“resource-constrained settings”). Much of the discussion of this study is navigating housing transitions considering the capacities of the households highlighted by the addition of the DVF framework. Thus, we emphasised the context in the title.

Title – “Optimising housing typology distributions for multi-hazard loss reductions in resource-constrained settings”

2. Abstract: The first sentence of the abstract could be more informative – it is not immediately clear what might be meant by the term “multi-hazard trade-offs”.

We have revised the statement to make it more descriptive, concisely however due to the word limit for abstracts.

Abstract – “Disaster loss estimations are valuable risk reduction tools but rarely consider the loss trade-offs when a building portfolio is subjected to multi-hazard impacts. [...]”

3. Line 103, page 3: I think you need to elaborate on why these might be insufficient- are you referring to insufficiency in terms of uptake or the strength of the measure itself?

We have clarified this statement to build upon the argument that the structural characteristics of typologies are often the main source of vulnerability, while repair and retrofit options can have limitations in achieving structural safety.

Introduction section > paragraph 5 – “[...] While DRR efforts have focused on the repair and retrofit of (existing) structural assets as practical solutions to strengthen physical assets, these measures can be insufficient considering that structural characteristics of typologies can govern vulnerability [27], [28].”

4. Line 106, page 3: Please specify the basis on which you are optimizing the building stock.

The basis to optimise the building stock is explained by the preceding sentence which mentions that addressing deeply seated vulnerabilities to multiple hazards requires more significant changes that can be achieved through changes to building typologies. We have reworded the next statement to explicitly

state that we then aim to optimise combinations of different building typologies in a heterogeneous building stock.

Introduction section > paragraph 5 – “[...] Thus, in a heterogeneous building stock, there is an opportunity to achieve optimal combinations of different building typologies to minimise multi-hazard impacts to safeguard the collective structural assets of a community.”

5. Line 150, page 4: Many readers may not be familiar with the national earthquake intensity scale. I would suggest the authors provide a brief explanation here of how it relates to peak ground acceleration.

We now have included a concise description about PEIS in this section of the manuscript.

Introduction section > paragraph 8 – “[...] PEIS range from Intensity I (scarcely perceptible; approximate peak ground acceleration (PGA) of < 0.0005 g; equivalent to Modified Mercalli Intensity (MMI) I) to Intensity X (completely devastating; approximate PGA of > 1.39 g; equivalent to MMI XII) (see [41], [42], [43], [44]). [...]”

6. Page 6: I appreciate that further details on the DVF framework are provided in the methods section at the end of the paper, but I think you need to make the reader aware of this here. As it stands, it is not immediately clear how a specific configuration is judged to be desirable, viable and feasible at this point in the text.

We now have revised this portion of the Introduction to briefly introduce the DVF framework, emphasising that the DVF analysis was done qualitatively to provide context in the discussion of potential housing stock transition. We have also mentioned that the DVF criteria were judged based on field study insights.

Introduction > paragraph 8 – “[...] Finally, acknowledging the socio-technical factors influencing residential construction in resource-constrained settings, the Pareto optimal solutions were contextualised against the desirability, viability, and feasibility (DVF) framework. We adopted this framework to qualitatively discuss whether potential solutions align with the households’ preferred mode of construction (“desirability”), suitable for the community in the long term (“viability”), and realistically achievable (“feasibility”). We assessed the DVF criteria based on field study insights of housing reconstruction trajectories of the households in Itbayat. For the entire data analysis procedures of this study, see “Methods” section.”

Responses to Reviewer Comments

We would like to thank the reviewers for the thoughtful comments and constructive suggestions which were valuable in improving the quality of this manuscript. We have carefully addressed each comment below. We list the review comments verbatim in bold, followed by our responses to these comments in plain text. We italicise with quotations key additions to the manuscript.

REVIEWER #4

I thank the authors for comprehensively responding to my original comments. I only have a few outstanding minor comments for the authors to address, which are all based on their responses to the first review:

1. Response to Comment 1a:

The authors provide a strong justification for the selection of hazards considered in their rebuttal letter, mentioning in-person consultations and the municipality's land use plan; this justification should also be incorporated into the text itself

We have now reflected in the revised manuscript the justifications on hazard selection as earlier stated in our rebuttal letter.

Introduction section > paragraph 7 – “[...] We focused on wind and seismic hazards because they are the most prominent hazards and the only known hazards that inflict significant damage to housing assets in the selected municipality. This was based on their Comprehensive Land Use Plan and on the in-person consultations in 2023 with the Municipal Planning and Development Office and the Municipal Engineering Office.”

2. Response to Comment 1b:

The authors should provide justification for how the performance-based design matrix is relevant to this context, especially given that the matrix was originally designed for earthquake hazards exclusively

The justification on the adaptability of the performance-based design matrix beyond seismic applications has now been added in the current version of the manuscript.

Methods section > Optimising direct economic losses > Selecting cases for Pareto optimal solution > paragraph 1 – “[...] While originally conceptualised for seismic applications, we extended the use of the performance-based design matrix to wind analysis. Given that the concept of the matrix generically establishes and maps (expected) performance levels of building/occupancy types against probability of hazard occurrences, it can then be adapted for other hazard contexts beyond earthquakes. For example, under “occasional” hazard intensities, a basic occupancy building (e.g., a residential dwelling) is recommended to meet life safety (or damage control) requirements – a consideration we deem is also applicable to other hazards, not just to seismic. Building codes provide regulations linking specific probabilistic hazard frequencies to required performance levels (e.g., life safety requirement for seismic hazards having 475-year return period). However, with emerging hazard projections coupled by the lived experiences of communities in disaster-affected regions, prescribed thresholds of hazard frequencies from building codes can be adapted in the performance-based design analysis to reflect both analytical and empirical sources of information.”

Responses to Reviewer Comments

We would like to thank the reviewers for the thoughtful comments and constructive suggestions which were valuable in improving the quality of this manuscript. We have carefully addressed each comment below. We list the review comments verbatim in bold, followed by our responses to these comments in plain text. We italicise with quotations key additions to the manuscript.

REVIEWER #4

I thank the authors for responding to my remaining comments. Overall, I am satisfied with the rebuttal. However, I would think that the justification for extending the application of the performance-based design matrix (in response to my second comment) could be slightly better weaved into the existing text. As it is, the justification mentions the concept of an occasional hazard intensity two paragraphs before it is explained; in my view, readability of the manuscript would be improved if the explanation was placed in the same paragraph. Perhaps the authors could first dedicate a paragraph to explaining and justifying the performance-based design matrix, including a description of the hazard intensity levels used. Then, the subsequent paragraphs could be dedicated to explaining and justifying the specific values of the hazard intensities used for each hazard in the case study.

As it currently stands, this section of the manuscript already follows the reviewer's recommended text flow in that a justification of the performance-based design matrix is presented first which is then immediately followed by a description of the hazard intensity levels used and capped off with a discussion of specific values of hazard intensities.

We understand, however, that the confusion might have come from the pre-introduction of the concept of occasional hazard intensities in the justification of the performance-based design matrix in the first paragraph. We would like to clarify that this text was brought up in reference to Figure 7 (which was already earlier introduced) to establish the relationship of hazard levels and the performance levels, and not to prematurely discuss the selection of occasional hazard intensities. To clear this out, we now have reworded the following sentence.

Methods section > Optimising direct economic losses > Selecting cases for Pareto optimal solution > paragraph 1 – “[...] *For example, under “occasional” hazard intensities as described in Figure 7, a basic occupancy building (e.g., a residential dwelling) is recommended to meet life safety (or damage control) requirements – a consideration we deem is also applicable to other hazards, not just to seismic.*”

Optimising building typology distributions in housing stocks reduces multi-hazard losses in resource-constrained communities

This study examines the multi-hazard risk associated with different configurations of housing types in a municipality of the Philippines. The authors determine configurations that would lead to lower multi-hazard losses than the current housing stock distribution. Furthermore, the best configurations are further analysed against a desirability, viability, and feasibility DVF framework to shed light on whether they could be implemented in practice. In summary, I believe this is an interesting contribution to the disaster engineering risk literature that would appeal to readers of this journal. I particularly welcome the authors' use of the DVF framework, which provides the study with some practical grounding. Having said this, I believe that changes to the manuscript are necessary before it could be deemed publishable, as detailed in the comments below.

Main comments:

1. One of my biggest concerns centers on the hazard component of the analysis, which needs to be substantially further justified and elaborated.
 - a. Firstly, the authors need to justify their focus on wind and earthquake hazard. What about other hazards that the Philippines is prone to, like floods and landslides? If this approach were to be replicated in other contexts, how should hazards be selected for consideration?
 - b. I urge the authors to reflect and further elaborate on the selection of specific hazard scenarios. How were the selected return periods deemed to be suitably representative of "occasional" versus "rare" classifications, why are just two return period pairs considered sufficient for this analysis (why not use a fully probabilistic hazard approach, for instance), how do they relate to any building codes implemented, and how are 0.2% and 1-5% probabilities of occurrence (in the case of the occasional classification) deemed to be compatible? In my opinion, the entire analysis lacks a strong rationale unless the previous questions can be clearly answered by the authors.
2. The results of the analysis are highly dependent on the fragility functions used. However, very little (if any) detail on these functions is provided by the authors. Please explain (for both hazards) (1) the applicability of the functions used for various typologies (including details on the data from which they are derived). This is especially important given the "heterogeneity of the housing stock" and "plurality of how households constructed their dwellings"; (2) the parameters of the functions used for each typology; and (3) the damage states captured in the fragility functions. Without these details, a judgement on the reliability of the study's outcomes cannot be made.
3. The discussion section requires some improvement, in my opinion:
 - a. The absence of hazard multipliers is not the only limitation of the hazard analysis; please revert to my first main comment
 - b. Line 421 page 11: don't these statements contradict the whole point of this study – i.e., the fact that we should not be considering hazards in isolation from each other?
 - c. The concept of resilience enhancement arises here (as well as in other sections of the text). However, enhancing resilience of the building stock requires much more than exclusive consideration of immediate economic losses – recovery time and recovery trajectories are significant components of resilience that are not dealt with in this study. Therefore, I do not think that the authors should be emphasizing the contribution of this study to resilience enhancement.
 - d. I acknowledge the authors' discussion on how the transition of the housing stock can be achieved. However, I struggle to understand how the "ideal ratio" would work in practice. Which households would be incentivized to achieve which typology, and how would the "ideal ratio" be consistently enforced?
 - e. The discussion on limitations ought to mention that the DVF framework has been calibrated based on secondary data

Technical comments on the methodology:

1. Why 11,628 scenarios? (I assume this has something to do with the 5% increments used, but please specify). Update: I see this is now clarified in Line 515, page 14 but I think the clarification should come earlier, where the 11,628 value is first introduced. I think you should also make it clear in the main text (on page 3) that an explanation for the number of scenarios considered is provided in the methods section.
2. Line 511-513, page 14: “This assumption is grounded on the observed heterogeneity of the present housing stock, implying that each of the typologies will likely be adopted or constructed by communities.” It is not clear how this statement is connected to the 5% assumption made.
3. Line 666: Why use the 95th percentile values – why not use the expected loss values, for instance? It would be useful to include a sensitivity analysis to understand if the results change depending on the loss statistic used.

More minor comments:

1. The title of the manuscript seems too obvious, i.e., if something is optimized (on the basis of losses), it is certain to lead to reduced losses. Furthermore, it is not clear why the analysis would be specifically restricted to “resource-constrained” communities.
2. Abstract: The first sentence of the abstract could be more informative – it is not immediately clear what might be meant by the term “multi-hazard trade-offs”.
3. Line 103, page 3: I think you need to elaborate on why these might be insufficient- are you referring to insufficiency in terms of uptake or the strength of the measure itself?
4. Line 106, page 3: Please specify the basis on which you are optimizing the building stock.
5. Line 150, page 4: Many readers may not be familiar with the national earthquake intensity scale. I would suggest the authors provide a brief explanation here of how it relates to peak ground acceleration.
6. Page 6: I appreciate that further details on the DVF framework are provided in the methods section at the end of the paper, but I think you need to make the reader aware of this here. As it stands, it is not immediately clear how a specific configuration is judged to be desirable, viable and feasible at this point in the text.